# RNF219 attenuates global mRNA decay through inhibition of CCR4-NOT complex-mediated deadenylation

Fabian Poetz [1,2], Joshua Corbo[3], Yevgen Levdansky [3], Alexander Spiegelhalter [1,2], Doris Lindner[1,2], Vera Magg[4], Svetlana Lebedeva [5], Jörg Schweiggert [2], Johanna Schott [1,2], Eugene Valkov [3✉] & Georg Stoecklin [1,2✉]

The CCR4-NOT complex acts as a central player in the control of mRNA turnover and mediates accelerated mRNA degradation upon HDAC inhibition. Here, we explored acetylation-induced changes in the composition of the CCR4-NOT complex by purification of the endogenously tagged scaffold subunit NOT1 and identified RNF219 as an acetylation-regulated cofactor. We demonstrate that RNF219 is an active RING-type E3 ligase which stably associates with CCR4-NOT via NOT9 through a short linear motif (SLiM) embedded within the C-terminal low-complexity region of RNF219. By using a reconstituted six-subunit human CCR4-NOT complex, we demonstrate that RNF219 inhibits deadenylation through the direct interaction of the α-helical SLiM with the NOT9 module. Transcriptome-wide mRNA half-life measurements reveal that RNF219 attenuates global mRNA turnover in cells, with differential requirement of its RING domain. Our results establish RNF219 as an inhibitor of CCR4-NOT-mediated deadenylation, whose loss upon HDAC inhibition contributes to accelerated mRNA turnover.

[1] Division of Biochemistry, Mannheim Institute for Innate Immunoscience (MI3), Medical Faculty Mannheim, Heidelberg University, 68167 Mannheim, Germany. [2] Center for Molecular Biology of Heidelberg University (ZMBH), German Cancer Research Center (DKFZ)-ZMBH Alliance, 69120 Heidelberg, Germany. [3] RNA Biology Laboratory, Center for Cancer Research, National Cancer Institute (NCI), Frederick, MD 21702-1201, USA. [4] Department of Infectious Diseases, Molecular Virology, Center for Integrative Infectious Disease Research (CIID), Heidelberg University, 69120 Heidelberg, Germany. [5] Berlin Institute for Molecular Systems Biology (BIMSB), Max Delbrück Center for Molecular Medicine, 10115 Berlin, Germany. ✉email: eugene.valkov@nih.gov; georg.stoecklin@medma.uni-heidelberg.de

Transcription and mRNA turnover jointly determine the abundance of any given mRNA in the cell, thus enabling adaptation of mRNA expression profiles with current cellular needs. Regulation of mRNA stability represents a key step in the control of post-transcriptional gene expression and shapes the kinetics of critical cellular processes such as cell proliferation, differentiation, immunity, and development[1–7]. Moreover, perturbation of mRNA turnover is associated with a variety of pathologies including oncogenesis, inflammation, and neurodegeneration[8–11]. Hence, uncovering molecular pathways governing both global and transcript-specific mRNA stability is essential to understanding the dynamics of gene expression.

The poly(A) tail at the 3′end of mRNAs, together with the 5′-m7G-cap structure at the 5′end, acts as an integral stability determinant and facilitates cap-dependent translation. In eukaryotes, deadenylation-dependent mRNA turnover is the primary pathway for the degradation of most mRNAs[12–15], whereby deadenylation represents the first and rate-limiting step in this process[16]. The multisubunit carbon catabolite repression 4 (CCR4)-negative on TATA-less (NOT) complex functions as the principal deadenylase responsible for processive poly(A) tail shortening in eukaryotic cells and is essential for normal rates of mRNA turnover[17–21].

The evolutionarily conserved CCR4-NOT complex contains two 3′−5′ exoribonucleases, the EEP-type deadenylase CCR4 and the DEDD-type deadenylase CCR4-associated factor 1 (CAF1). The concerted action of both 3′−5′ exoribonucleases contributes to the shortening of poly(A) tails in yeast and human cells, albeit with different specificities towards poly(A)-binding protein (PABP)-protected segments[22,23]. In contrast to the yeast CCR4-NOT complex, the composition of the metazoan complex is heterogenous since there are two paralogs each of CAF1 (CAF1a/CNOT7, CAF1b/CNOT8) and CCR4 (CCR4a/CNOT6, CCR4b/CNOT6L), whose incorporation into the complex is mutually exclusive[24]. The complex is organized around NOT1, which serves as a large scaffold protein that provides a binding platform for the catalytic deadenylase module and three non-catalytic modules[25]. CCR4 stably associates with the complex via CAF1, which docks onto the central, α-helical MIF4G domain of NOT1, together forming the catalytic module[26,27]. The C-terminal end of NOT1 interacts with the NOT2 and NOT3 subunits to form the so called NOT module[28,29]. The N-terminal portion of NOT1 stably binds the metazoan-specific NOT10 and NOT11 subunits[30,31]. The third non-catalytic module assembles on a central region of NOT1 close to the catalytic module, and includes the NOT9 subunit, the human homolog of yeast Caf40[32,33]. The RING-type E3 ligase NOT4 interacts via NOT9/Caf40 with the CCR4-NOT complex, though it is stably associated with the complex only in yeast[24,34–38]. In metazoans, NOT9 serves as an adaptor for multiple RNA-binding proteins (RBPs) such as Tristetraprolin (TTP), GW182/TNRC6, Bag-of-marbles (Bam), and Roquin which induce selective mRNA degradation[32,33,39–41].

Aside from recruitment of specific target mRNAs to the CCR4-NOT complex via RBPs recognizing 3′UTR elements[2,42–48] the CCR4-NOT complex also promotes the deadenylation of bulk mRNA through a generic pathway that involves PABPC1 and members of the BTG/TOB family[49–51]. Moreover, CCR4-NOT-mediated deadenylation is regulated by post-translational modifications (PTMs) such as ubiquitination and acetylation of CAF1[52,53] as well as phosphorylation of TOB2[54]. However, little is known about signaling-dependent alterations in the composition of the CCR4-NOT complex and how these affect its enzymatic activity. Based on our previous observation that HDAC inhibition dramatically accelerates global poly(A) RNA turnover via CCR4-NOT-mediated deadenylation[53], we set out to explore whether pharmacological HDAC inhibition alters the composition of the CCR4-NOT complex.

## Results

**RNF219 is an acetylation-regulated subunit of the CCR4-NOT complex.** We previously discovered that protein acetylation has a profound impact on global mRNA turnover[53], whereby inhibition of HDAC1 and HDAC2 induces widespread degradation of bulk poly(A) RNA in mammalian and *Drosophila melanogaster* (*Dm*) cells. Since acetylation-induced mRNA turnover depends on deadenylation[53], we asked whether there may be acetylation-regulated changes in the protein interactome of the CCR4-NOT complex. So far, the composition of the mammalian CCR4-NOT complex was assessed by affinity purification-mass spectrometry (MS) using ectopically expressed subunits[24,31,55]. Since we wanted to explore physiological changes in the composition of the CCR4-NOT complex, we employed a genome editing approach to purify the endogenous complex from HeLa cells. To this end, the start codon of the scaffold subunit NOT1, located in exon 2, was targeted by CRISPR/Cas9-mediated cleavage, followed by insertion of an N-terminal 3xFLAG-2xStrep (FST) tag via homology-directed repair (Fig. 1a). Individual HeLa clones were tested for FST-NOT1 expression, and Sanger sequencing of the targeted locus in clone #47, which was used for all subsequent experiments, confirmed in-frame integration of the FST sequence downstream of the start codon in one *CNOT1* allele (Supplementary Fig. 1). FST-NOT1 co-sediments with other CCR4-NOT subunits in a linear glycerol gradient (Supplementary Fig. 2a), indicative of proper incorporation and assembly of the endogenous complex. Further characterization of clone #47 revealed that the expression level of NOT1 is similar to that of parental HeLa cells (Supplementary Fig. 2b). Moreover, treatment of clone #47 with the class I-specific HDAC inhibitor Romidepsin (RMD)[56] led to a more than two-fold acceleration of poly(A) RNA turnover (Supplementary Fig. 2c; half-life of 7.6 ± 1.1 vs. 3.6 ± 0.5 h), similar to what was observed in parental HeLa cells[53].

We then purified the complex by FLAG-based immunoprecipitation (IP) of FST-NOT1, and analyzed its composition by protein MS. As a negative control, the same IP was conducted in parental HeLa cells. Thereby we observed the specific enrichment of all canonical subunits of the CCR4-NOT complex in the FST-NOT1 purification, while none of these proteins were detected in the control IP (Fig. 1b, blue dots, Supplementary Data 1). Our purification strategy further recovered the mammalian-specific component TNKS1BP1 (also known as TAB182), which was identified as a stable interactor of the CCR4-NOT complex in previous studies[24,57,58]. Six additional proteins were strongly enriched in our FST-NOT1 purification (Fig. 1b, red dots): four and a half LIM domains protein 2 (FHL2), ring finger protein 219 (RNF219), two actin-associated proteins CAPZA1 and CAPZB, ribonucleoprotein PTB-binding 1 (RAVER1), as well as ribosomal protein L7 (RPL7). Purification of FST-NOT1 followed by western blot analysis confirmed that RNF219, FHL2 as well as CAPZA1 and CAPZB associate with the CCR4-NOT complex, whereas RAVER1 and RPL7 could not be detected in the IP (Fig. 1c). Importantly, the co-purification of both RNF219 and FHL2 with the CCR4-NOT complex was not affected by RNase treatment (Supplementary Fig. 2d, e), indicating that their association is based on protein–protein interactions. Moreover, well-known interactors of the CCR4-NOT complex such as members of the BTG/TOB family[59,60], the facultative cofactor NOT4[24,37] or other decay-promoting RBPs were not enriched by FST-NOT1 purification (Fig. 1b), suggesting that our isolation strategy retained only factors that are stably associated with the CCR4-NOT complex.

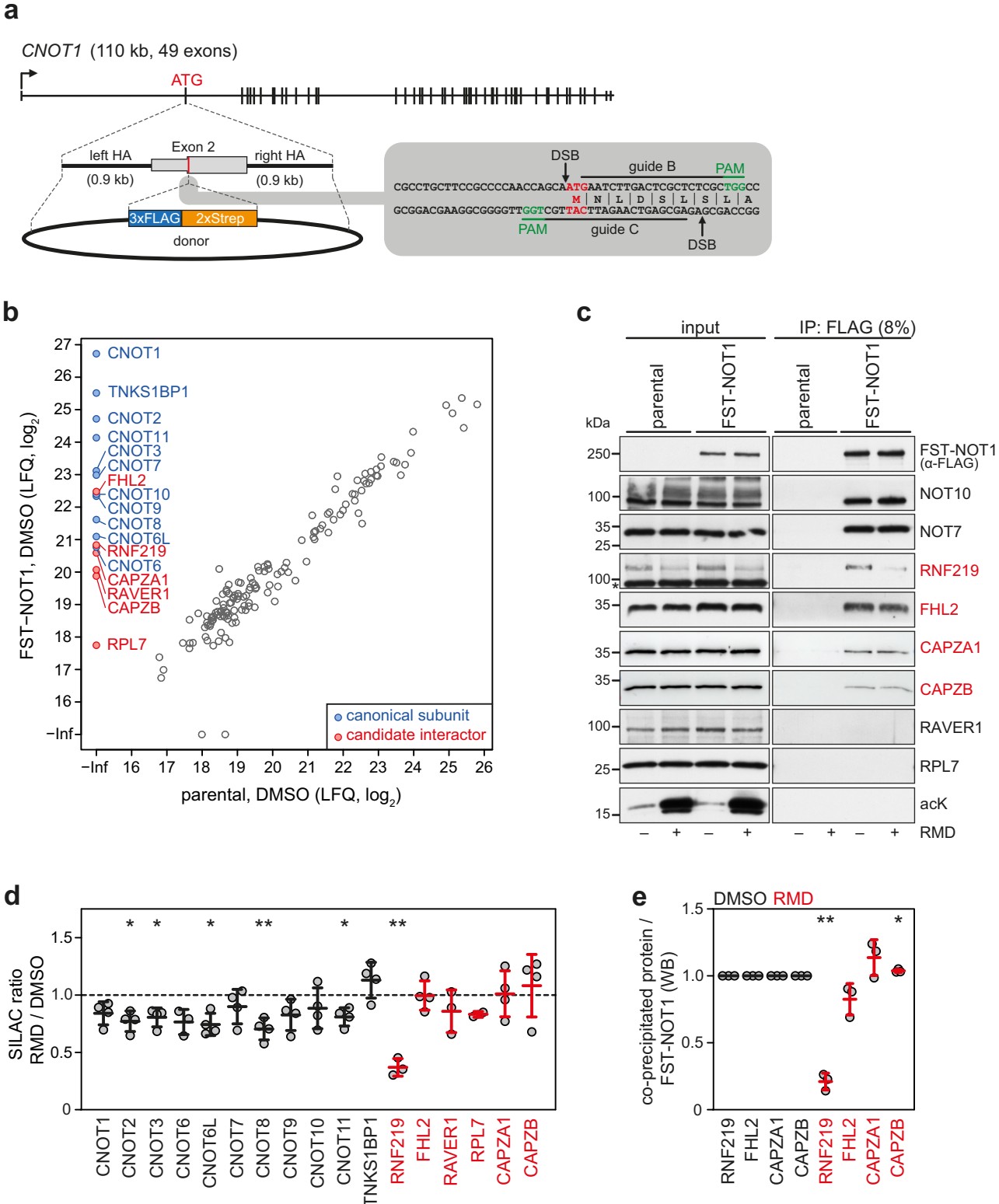

To investigate the effect of pharmacological HDAC inhibition on the composition of the CCR4-NOT complex, we made use of quantitative proteomics by stable isotope labeling with amino acids in cell culture (SILAC; Supplementary Fig. 2f). Following differential isotope labeling of FST-NOT1 HeLa cells treated with RMD or solvent (DMSO), equal quantities of heavy- and light-labeled protein extract were combined, subjected to FST-NOT1 purification, and analyzed by protein MS (Supplementary Data 1).

The same approach was also used to assess background binding in parental HeLa cells. Whereas the association of all canonical subunits with FST-NOT1 was barely affected by RMD treatment, the amount of co-precipitated RNF219 was reduced by more than two-fold (Fig. 1d), indicating that this factor may be regulated by acetylation. Consistent with the SILAC-based quantification, the amount of co-precipitated RNF219 was reduced by approximately four-fold upon RMD treatment when assessed by western

**Fig. 1 Purification of the endogenously tagged CCR4-NOT complex. a** Schematic illustration of the CRISPR/Cas9 strategy for N-terminal tagging of endogenous *CNOT1* in HeLa cells. Integration of the 3xFLAG-2xStrep (FST) sequence downstream of the ATG start codon was achieved by two guide RNAs introducing double-strand DNA breaks (DSBs) and homology-directed repair using a donor plasmid; HA, homology arm; PAM, protospacer adjacent motif. **b** Scatter plot depicts proteins identified by mass spectrometry following FLAG IP from FST-NOT1 or parental HeLa cells as negative control. Canonical subunits of the CCR4-NOT complex are shown in blue, candidate interactors in red; values represent the means of $\log_2$-transformed LFQ intensities of $n = 4$ independent biological replicates; LFQ, label-free quantification. **c** Validation of candidate CCR4-NOT interactors by western blot analysis using antibodies as indicated. The endogenous CCR4-NOT complex was purified by FLAG IP from FST-NOT1 or parental HeLa cells treated with 20 nM RMD or an equal volume of DMSO for 16 h. **d** Dot plot depicts the SILAC ratio of canonical CCR4-NOT subunits and candidate interactors determined by mass spectrometry. Samples were processed as illustrated in Supplementary Fig. 2f; values are presented as mean ± SD ($n \geq 2$) including two label-swap replications (CNOT2, $n = 4$, $p = 0.0147$; CNOT3, $n = 4$, $p = 0.018$; CNOT6L, $n = 4$, $p = 0.0133$; CNOT8, $n = 4$, $p = 0.0084$; CNOT11, $n = 4$, $p = 0.0173$; RNF219, $n = 3$, $p = 0.0049$). **e** Dot plot shows the quantification of western blots as in (**c**). Values are presented as mean ± SD ($n = 3$; RNF219, $p = 0.0021$; CAPZB, $p = 0.0326$). Statistical significance in panel (**d**) and (**e**) was determined by using a two-sided, one-sample $t$ test ($^*p < 0.05$; $^{**}p < 0.01$). Source data for panels (**b–e**) are provided as a Source Data file.

blot analysis, whereas the association of FHL2 as well as CAPZA1 and CAPZB remained largely unaffected (Fig. 1c, e).

We then monitored association of RNF219 and FHL2 with the CCR4-NOT complex by glycerol gradient centrifugation and found that both proteins co-sediment together with core components of the complex in fractions 8–12 (Supplementary Fig. 3a, b). Moreover, both proteins shifted to lighter fractions upon siRNA-mediated knock-down (KD) of the scaffold component NOT1 (Supplementary Fig. 3a, b). Co-IP of RNF219 and FHL2 with FST-NOT1 was also observed when isolated from heavy fractions of a glycerol gradient (Supplementary Fig. 3c), providing further evidence that both factors are stably associated with the core CCR4-NOT complex.

To better understand the reason for reduced association of RNF219 with the CCR4-NOT complex, we measured changes in *RNF219* pre-mRNA, mRNA, and protein expression upon HDAC inhibition by RMD and two pan-HDAC inhibitors, Trichostatin A (TSA) and suberoylanilide hydroxamic acid (SAHA). This analysis revealed a significant reduction of *RNF219* pre-mRNA, mRNA, and protein expression after 16 h of inhibition (Supplementary Fig. 4a, b). The observation that all three HDAC inhibitors decreased *RNF219* pre-mRNA levels (Supplementary Fig. 4a, right panel) prompted us to further investigate the involvement of HDACs in RNF219 transcription. Analysis of ChIP-Seq data derived from primary human CD4[+] T-cells[61] revealed HDAC1 and RNA polymerase II (PolII) occupancy across the first exon of *RNF219* (Supplementary Fig. 4c), suggesting a function of HDAC1 in RNF219 transcription. We therefore tested siRNA-mediated knock-down of HDAC1 and found this to significantly reduce *RNF219* pre-mRNA expression (Supplementary Fig. 4d, right panel) and cause a moderate decrease in *RNF219* mRNA and protein levels (Supplementary Fig. 4d, e). Hence, HDAC1 inhibition compromises RNF219 expression due to transcriptional downregulation. Although the inhibition of class-I HDACs accelerates the degradation of poly(A)-containing mRNAs[53], *RNF219* mRNA stability was barely affected by RMD treatment (Supplementary Fig. 4f), excluding the possibility that reduced RNF219 expression is linked to changes in its mRNA half-life. In conclusion, transcriptional regulation represents the primary reason for the RMD-mediated decrease in RNF219 expression, which ultimately results in the depletion of RNF219 from the CCR4-NOT complex.

Taken together, purification of the endogenous complex identified RNF219, FHL2 as well as CAPZA1 and CAPZB as stable interactors of the CCR4-NOT deadenylase. While the association of FHL2, CAPZA1 and CAPZB is acetylation-independent, RNF219 dissociates from the CCR4-NOT complex as a consequence of its reduced expression upon RMD-mediated hyperacetylation.

**RNF219 binds to the NOT9 module via a short linear motif**. So far, RNF219 was characterized as a ubiquitin ligase that promotes replication origin firing via ubiquitination of the origin recognition complex[62]. Moreover, an interactome study indicated a connection between RNF219 and the CCR4-NOT complex, whereby RNF219 was placed in close proximity to NOT9 based on biotinylation experiments[55]. Guénolé et al. recently reported that residues 540–549 in the C-terminus of RNF219 mediate binding to the CCR4-NOT complex (Fig. 2a)[63]. Interestingly, several RBPs were previously shown to associate with the CCR4-NOT complex through direct binding to the NOT9/Caf40 subunit via short linear motifs (SLiMs), also termed Caf40-binding motifs, which are embedded within low-complexity regions[37,40,41].

We first investigated whether RNF219 may associate with the CCR4-NOT complex via the NOT9 module. Strikingly, a partial KD of NOT9 was sufficient to prevent the co-precipitation of RNF219, but not FHL2, with FST-NOT1 in HeLa cells (Fig. 2b), indicating that NOT9 is essential for recruitment of RNF219 to the CCR4-NOT complex.

To test whether this interaction is direct, we produced a recombinant fragment comprising the entire C-terminal low-complexity region of RNF219 (residues 434-726), termed RNF219-C (Fig. 2a). The fragment was tagged at the N-terminus with maltose-binding protein (MBP) and immobilized on StrepTactin beads via a C-terminal StrepII tag. RNF219-C was then incubated with all four functional CCR4-NOT subcomplexes reconstituted from purified recombinant proteins[25], and found to specifically bind to the NOT9 module, but not the NOT10/11, the catalytic NOT6/7, or the NOT module (Fig. 2c).

Next, we sought to confirm that residues 540-549 represent a SLiM through which RNF219 associates with the CCR4-NOT complex and expressed RNF219 fused N-terminally to a 3xFLAG-SBP (3xFS) tag in HeLa cells. Whereas wild-type (WT) RNF219 efficiently co-immunoprecipitated NOT9 as well as NOT2, the ΔSLiM mutant failed to do so (Fig. 2d). This result was corroborated by a reciprocal pull-down where HA-tagged NOT9 was found to co-immunoprecipitate WT RNF219 but not the ΔSLiM mutant (Supplementary Fig. 5). Furthermore, we observed that RNF219-ΔSLiM co-sediments in lighter fractions of a linear glycerol gradient as compared to its WT counterpart (Fig. 2e, quantification on the right side). In fact, RNF219-ΔSLiM showed only a small shift by one fraction in the glycerol gradient, indicating that apart from the C-terminal SLiM, RNF219 might establish additional, low-affinity interactions with the CCR4-NOT complex.

Taken together, our interaction analyses provide compelling evidence for a stable and specific interaction between RNF219 and the NOT9 module of the CCR4-NOT complex, mediated by a SLiM embedded within the low-complexity region of RNF219.

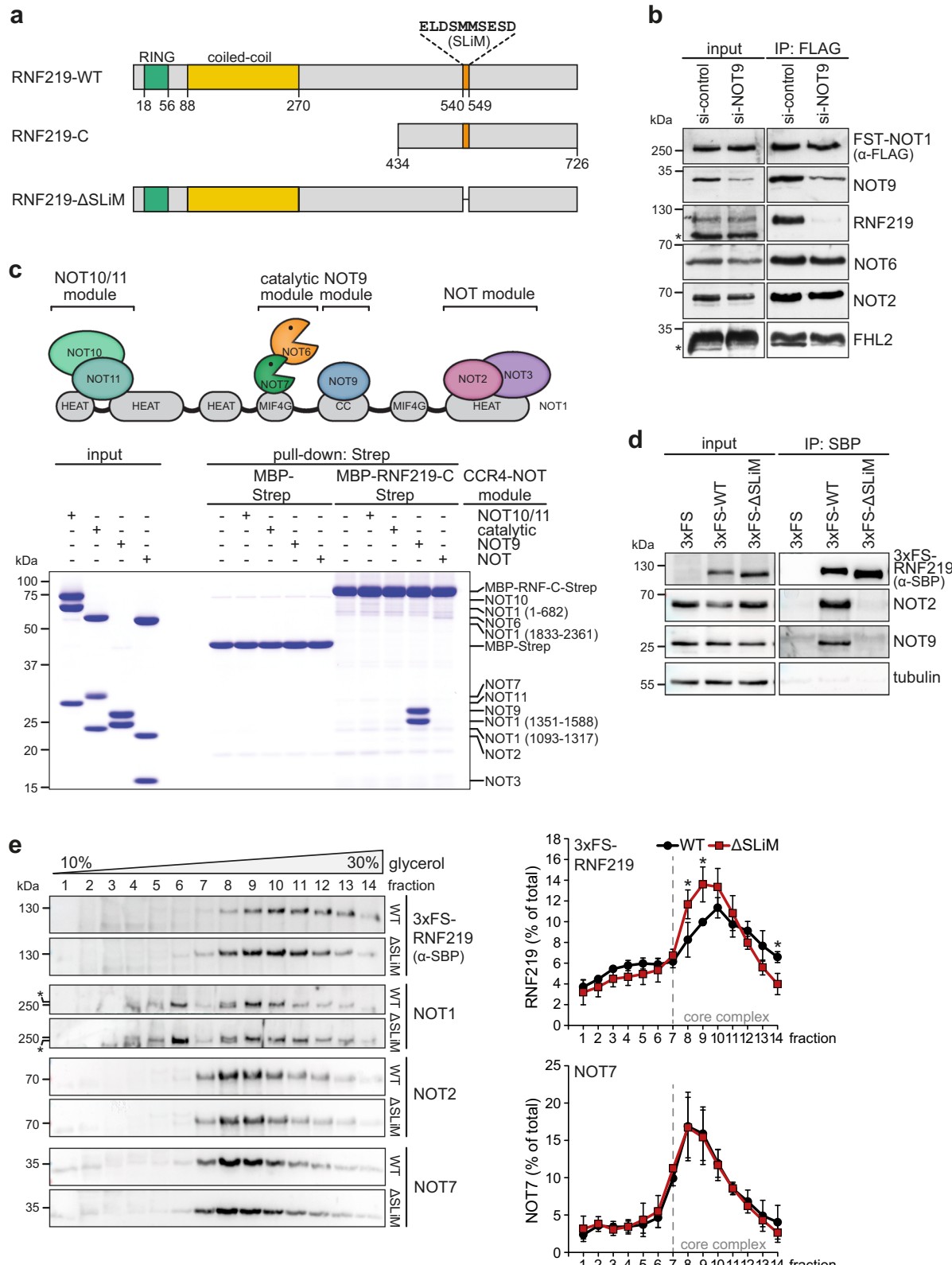

**RNF219 is a CCR4-NOT-associated active ubiquitin ligase.** RNF219 harbors an N-terminal RING-type E3 ligase domain, which displays high amino acid sequence conservation among vertebrates (Supplementary Fig. 6a) and conforms to the consensus sequence of other RING-type E3 ligases (Fig. 3a)[64]. Indeed, two recent studies independently confirmed that RNF219 is an active E3 ubiquitin ligase[62,63].

To determine if RNF219 acts as an E3 ubiquitin ligase within the CCR4-NOT complex, we made use of the observation that E3 ligases frequently undergo autoubiquitination[65]. We assessed the protein half-life of RNF219 following arrest of protein synthesis by treatment with cycloheximide (CHX) and found that RNF219-WT is rapidly degraded with a half-life of 2.5 h (Fig. 3b). In contrast, mutation of two conserved cysteines within the RING

**Fig. 2 RNF219 associates with the CCR4-NOT complex via NOT9. a** Schematic illustration of the domain architecture of RNF219; RING, really interesting new gene; SLiM, short linear motif. RNF219-C (amino acid residues 434–726) and RNF219-ΔSLiM are shown below. **b** Co-precipitation of RNF219 was examined by FLAG IP of the endogenous CCR4-NOT complex from FST-NOT1 HeLa cells following transfection with either control or NOT9-targeting siRNAs for 48 h. Proteins were analyzed by western blotting with antibodies as indicated; the asterisks denote non-specific bands. **c** Coomassie-stained polyacrylamide gel of in vitro pull-down assays with recombinant RNF219-C fused to maltose-binding protein (MBP) and a StrepII (Strep) affinity tag upon incubation with four different CCR4-NOT modules as indicated. **d** Co-IP of endogenous NOT2 and NOT9 with RNF219 using HeLa cells stably expressing 3xFLAG-SBP (3xFS)-RNF219-WT or 3xFS-RNF219-ΔSLiM. Precipitated proteins were detected by western blot analysis using antibodies as indicated. **e** Co-sedimentation analysis of CCR4-NOT subunits by 10–30% linear glycerol gradient fractionation and western blotting from HeLa cells stably expressing 3xFS-RNF219-WT or 3xFS-RNF219-ΔSLiM. The core CCR4-NOT complex elutes in fractions 7–14. Data from one representative experiment are shown (left panel), and the relative distribution of RNF219 and NOT7 along the gradient was quantified (right panel); values are presented as mean ± SD (WT $n = 3$; ΔSLiM $n = 4$; fraction 8, $p = 0.0313$; fraction 9, $p = 0.0154$; fraction 14, $p = 0.0106$). The asterisks denote non-specific bands. $p$ values were calculated using a two-tailed, paired $t$ test (*$p < 0.05$). Source data for panels (**b–e**) are provided as a Source Data file.

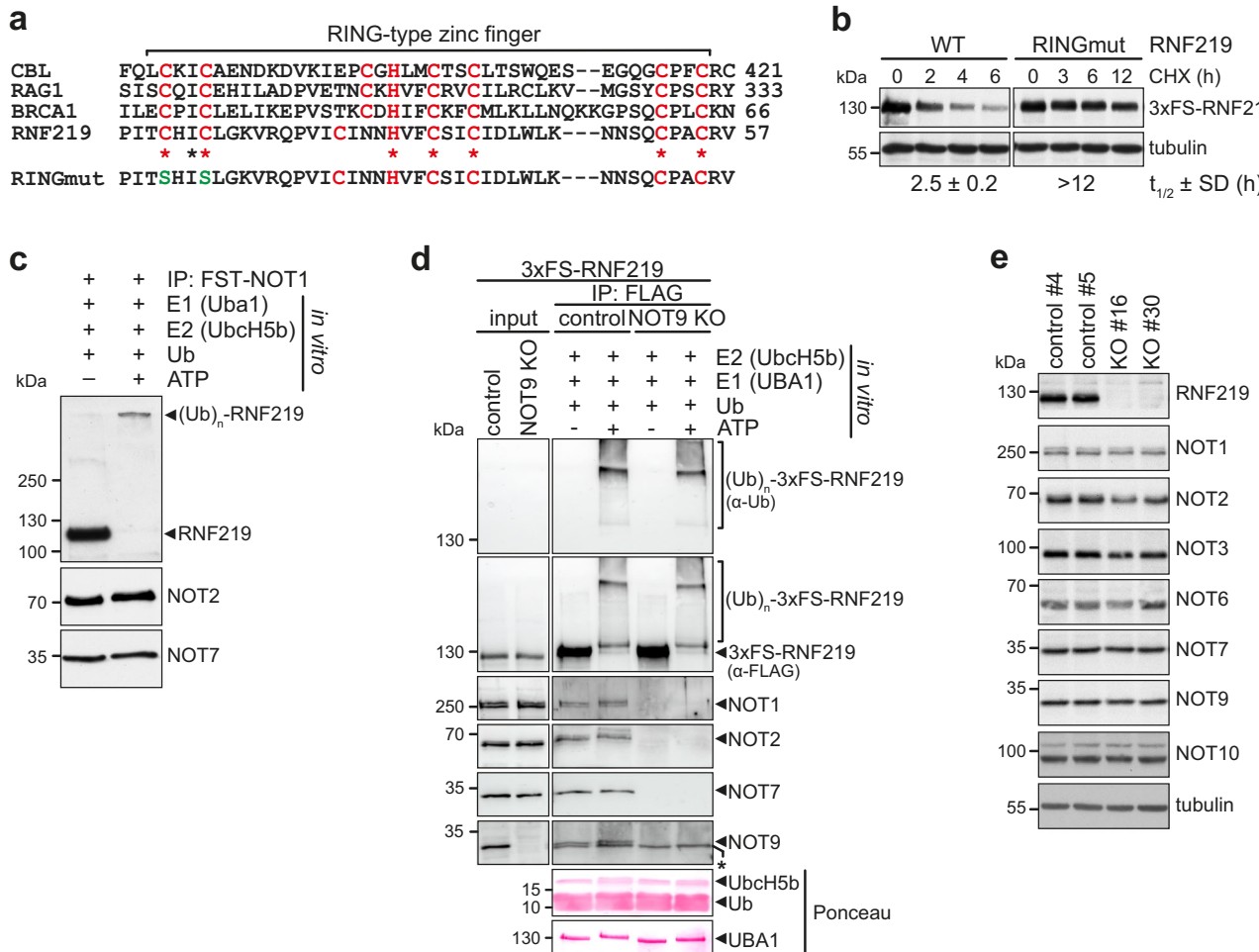

**Fig. 3 RNF219 is a CCR4-NOT complex-associated RING-type E3 ligase. a** Amino acid sequence alignment of the RNF219 RING domain with other RING-type E3 ligases. Cysteine and histidine residues involved in zinc atom coordination are highlighted in red. Cysteines at position 18 and 21 were mutated to serine in RNF219-RINGmut, shown in green. **b** Protein stability of 3xFS-RNF219-WT and 3xFS-RNF219-RINGmut was assessed in transiently transfected HeLa cells following translation shut-off with 100 μM cycloheximide (CHX). Proteins were isolated at regular time intervals and detected by western blot analysis using α-FLAG and α-tubulin antibodies. 3xFS-RNF219-WT and -RINGmut protein half-lives are presented as mean ± SD ($n = 3$). **c** In vitro ubiquitination assay showing autoubiquitination of RNF219 within the endogenous CCR4-NOT complex. Native CCR4-NOT complexes were purified from HeLa cells expressing FST-NOT1, eluted with 3xFLAG peptide, and incubated with recombinant E1 enzyme (UBA1), E2 enzyme (UbcH5b) and ubiquitin in the absence or presence of 5 mM ATP. Reaction products were detected by western blot analysis as indicated. **d** In vitro ubiquitination assay showing autoubiquitination of RNF219 in the presence and absence of the CCR4-NOT complex. CCR4-NOT complexes were purified from CRISPR control or NOT9 KO HeLa cells stably expressing 3xFS-RNF219 by FLAG IP and used for in vitro ubiquitination assays as described in (**c**). The asterisk denotes a non-specific band. **e** Total protein lysates derived from two control and two RNF219 KO HeLa clones were analyzed by western blotting using antibodies as indicated. Source data for panels (**b–e**) are provided as a Source Data file.

domain (RNF219-RINGmut; Fig. 3a) lead to an increase in protein stability with a half-life exceeding 12 h (Fig. 3b), suggesting that RNF219 undergoes rapid turnover due to autoubiquitination.

To assess RNF219 E3 ligase activity more directly, we performed in vitro ubiquitination assays with 3xFS-tagged RNF219 immuno-purified from HEK293 cells. 3xFS-RNF219 was then incubated with recombinant E1 enzyme (UBA1), different E2 enzymes (UbcH5b/c, UbcH10, Cdc34) and ubiquitin in the absence or presence of ATP (Supplementary Fig. 6b). During the reaction, RNF219-WT was extended to a range of high molecular weight species, indicative of E3 ligase autoubiquitination, whereas RNF219-RINGmut did not show this activity (Supplementary Fig. 6b). Moreover, these in vitro ubiquitination assays revealed that RNF219 cooperates with the promiscuous ubiquitin-conjugating enzymes UbcH5b and UbcH5c[66], but not with the cell cycle-related UbcH10 or Cdc34 enzymes[67].

Because of its stable association with the CCR4-NOT complex, we then asked if RNF219 retains E3 ligase activity when bound to the deadenylase complex. In vitro RNF219 autoubiquitination assays were carried out with native CCR4-NOT complexes isolated from FST-NOT1 HeLa cells. Indeed, all the CCR4-NOT-bound RNF219 protein showed extensive autoubiquitination as evident from a drastic shift in molecular weight (Fig. 3c). To assess if the association of RNF219 with the CCR4-NOT complex affects its E3 ligase activity, we generated NOT9 knock-out (KO) HeLa cells stably expressing 3xFS-RNF219 by CRISPR/Cas9-mediated genome editing (Supplementary Fig. 7). Characterization of a selected NOT9 KO clone confirmed that it expresses equal levels of 3xFS-RNF219 as compared to a CRISPR control clone, and lacks NOT9 expression (Fig. 3d, input) due to deletions downstream of the guide RNA target sequence in exon 2 (Supplementary Fig. 8a, b). As expected from our in vitro binding results (Fig. 2c), loss of NOT9 expression precluded co-purification of the CCR4-NOT complex by 3xFS-RNF219 IP (Fig. 3d). Interestingly, in vitro ubiquitination assays performed with 3xFS-RNF219 IP eluates derived from control or NOT9 KO cells did not show differences in RNF219 autoubiquitination. Hence, RNF219 is an active CCR4-NOT-associated RING-type E3 ligase, whose activity is not affected by its binding to the CCR4-NOT complex.

Given its capacity to promote its own degradation (Fig. 3b) and act as an efficient ubiquitin E3 ligase within the CCR4-NOT complex (Fig. 3c, d), we asked if RNF219-mediated ubiquitination may affect the abundance of CCR4-NOT subunits or alter the composition of the complex. To this end, we generated RNF219 KO HeLa cells by CRISPR/Cas9-mediated genome editing (Supplementary Fig. 9a). Characterization of two selected RNF219 KO clones confirmed that they fail to express RNF219 (Fig. 3e) due to a deletion (clone #16) or insertion (clone #30) in the vicinity of the ATG start codon (Supplementary Fig. 9b). However, the abundance of core CCR4-NOT subunits (NOT1-3, NOT6-7, NOT9-10) remained unaffected in both RNF219 KO clones (Fig. 3e). Moreover, the loss of RNF219 did not lead to detectable differences in the sedimentation of core CCR4-NOT subunits (NOT1-3, NOT6-7) in a linear glycerol gradient (Supplementary Fig. 10). These results indicate that RNF219-mediated ubiquitination does not influence the composition of the CCR4-NOT complex or promote proteolytic degradation of core subunits other than RNF219 itself.

**RNF219 inhibits CCR4-NOT-mediated deadenylation.** To delineate the function of RNF219 within the CCR4-NOT complex, we first analyzed its impact on CCR4-NOT-mediated deadenylation in vitro. Previously, we showed that the human six-subunit CCR4-NOT$_{MINI}$ complex (Fig. 4a), which contains the full-length exoribonucleases NOT6 (CCR4a) and NOT7 (CAF1a), the C-terminal half of NOT1 (residues 1093-2376), NOT9 (residues 19-285), as well as the minimal constructs of NOT2 (residues 344-540) and NOT3 (residues 607-753), exhibits almost the same deadenylation activity as the full complex[25]. Since the C-terminal half of RNF219 efficiently binds to the NOT9 module (Fig. 2c), we assessed the impact of RNF219-C on the activity of the reconstituted CCR4-NOT$_{MINI}$ complex by monitoring the deadenylation of a 5′-fluorescein-labeled 7-mer RNA substrate containing a poly(A) tail of twenty adenosines. Addition of an almost saturating concentration of MBP-tagged RNF219-C, equivalent to a 45-fold molar excess, led to pronounced inhibition of deadenylation (Fig. 4b, compare MBP-Strep alone to MBP-RNF219-C). The extent of inhibition was similar to that observed for the Caf40-binding motif of the *Dm* Bam protein (Fig. 4b), a known inhibitor of CCR4-NOT-mediated deadenylation[25]. To quantify the kinetics of in vitro deadenylation[68], line scans were used to identify the most abundant deadenylation intermediates (Supplementary Fig. 11a), from which an apparent deadenylation rate was calculated (Fig. 4c). This analysis revealed that RNF219-C elicited a three- to four-fold reduction in the apparent deadenylation rate of the CCR4-NOT$_{MINI}$ complex. Moreover, RNF219-mediated inhibition of deadenylation is a consequence of its direct association with the NOT9 module since no change in the apparent deadenylation rate was detected in the presence of the nuclease module alone (Fig. 4d, e). A titration experiment revealed that CCR4-NOT$_{MINI}$ was progressively inhibited by increasing concentrations of RNF219-C with a 50-fold molar excess (2.5 μM) being sufficient to almost completely inhibit deadenylation (Fig. 4f). RNF219 acts as an efficient antagonist of CCR4-NOT$_{MINI}$-mediated deadenylation since inhibition was already apparent at an equimolar concentration of substrate, CCR4-NOT$_{MINI}$ and RNF219-C (0.05 μM). Importantly, RNF219-C did not bind to the RNA substrate at any of the tested concentrations (Supplementary Fig. 11b, c), excluding the possibility that the observed inhibition of deadenylation is due to RNA sequestration. Hence, RNF219 has the capacity to directly inhibit CCR4-NOT-mediated deadenylation in vitro. Given that RNF219 interacts with the CCR4-NOT complex through the NOT9 module (Fig. 2c), this result is consistent with our previous finding that NOT9 serves as the dominant RNA-binding site of the CCR4-NOT complex in vitro[25].

**The SLiM of RNF219 folds into an α-helix on the RNA-binding surface of NOT9.** To characterize the interaction of the RNF219 SLiM with NOT9 on a structural level, we employed the AlphaFold2 structure prediction tool[69,70]. A single polypeptide comprising the NOT9 sequence fused C-terminally to the RNF219 disordered C-terminal region via a polyglycine linker was provided to the prediction analysis pipeline in absence of any imposed constraints, and five models were generated (Supplementary Fig. 12a). Remarkably, in each of the five models, the RNF219 SLiM was predicted to fold into an amphipathic α-helix extending through residues S525–S548 and is positioned across the concave surface of NOT9 (Fig. 5a, Supplementary Fig. 12a), which consists of α-helical armadillo repeats[32,33,71]. The SLiM binds into the positively charged pocket of NOT9 (Fig. 5b) that was proposed as a nucleic acid-binding groove[71], and thus occupies the same binding surface as the previously described SLiMs of *Dm* Roquin[40] and Bam[41] (Supplementary Fig. 12b).

We then examined possible contacts between the RNF219 SLiM and the positively charged pocket in NOT9 and noted three side chains of the hydrophobic residues L541, M544, and M545 in

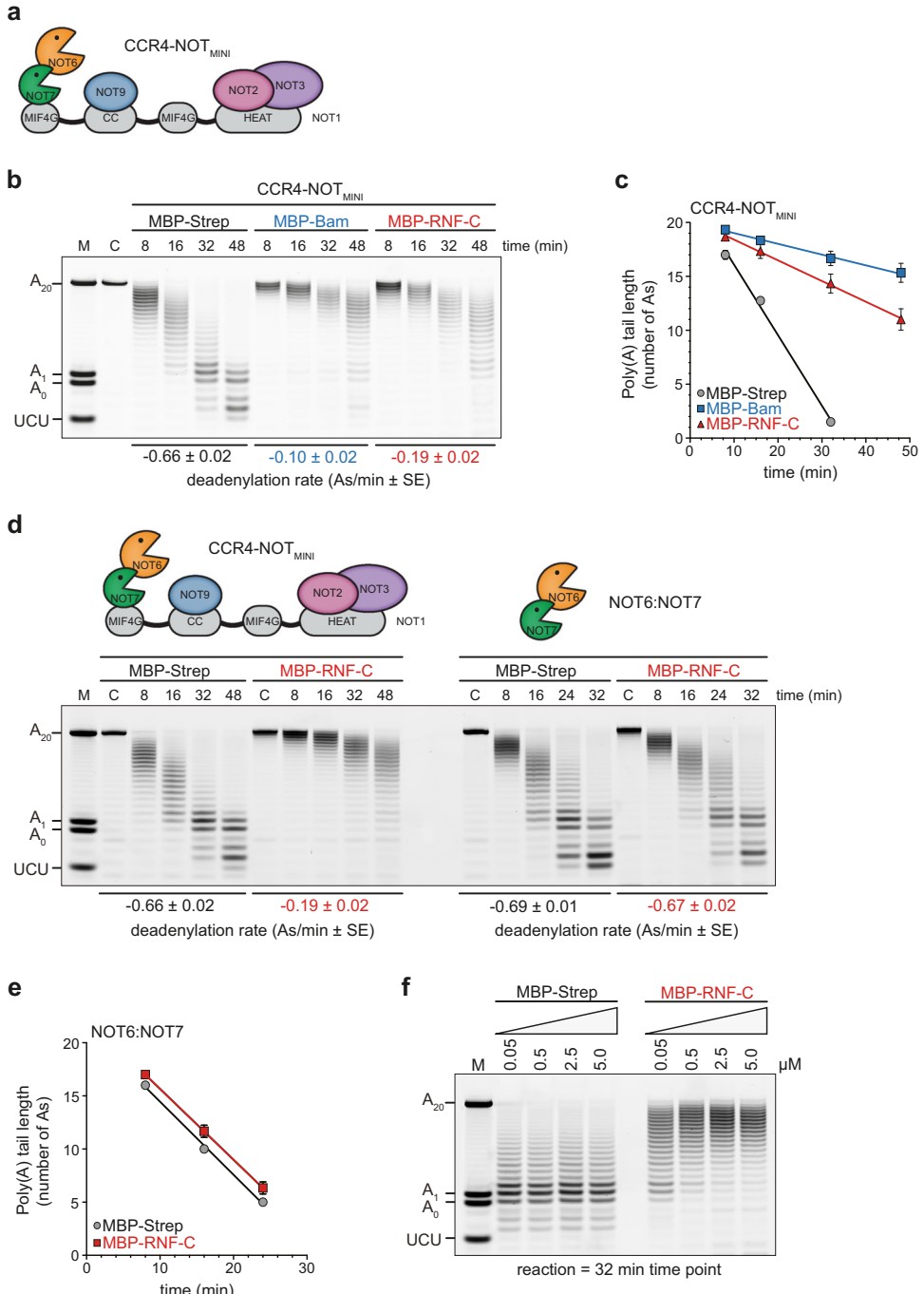

**Fig. 4 RNF219 inhibits CCR4-NOT-mediated deadenylation in vitro. a** Scheme representing the architecture of the recombinant CCR4-NOT$_{MINI}$ complex; MIF4G, middle domain of eukaryotic initiation factor 4G; CC, coiled-coil. **b** In vitro deadenylation assay with 50 nM of a 7-mer-A$_{20}$ RNA substrate, 50 nM CCR4-NOT$_{MINI}$ complex, and 2250 nM of either MBP-Strep, *Dm* MBP-Bam Caf40-binding motif (residues 13-36) or MBP-RNF219-C. Reactions were stopped at the indicated time points and analyzed by electrophoresis using a 20% denaturing polyacrylamide gel; M, RNA size marker; C, RNA alone control. **c** Quantification of the change in poly(A) tail length in (**b**) by plotting the most abundant tail length at each time point. Linear regression was used to determine the apparent deadenylation rate (As/min); values are presented as mean ± SE ($n = 3$). **d** In vitro deadenylation assay with 50 nM of a 7-mer-A$_{20}$ RNA substrate, 50 nM CCR4-NOT$_{MINI}$ complex (left) or 250 nM NOT6:NOT7 exonuclease heterodimer (right), and 2250 nM of MBP-Strep or MBP-RNF219-C. Reactions were stopped at the indicated time points and analyzed as described in (**b**). **e** Quantification of the change in poly(A) tail length in (**d**) by plotting the most abundant tail length at each time point; values are presented as mean ± SE ($n = 3$). **f** In vitro deadenylation assay with 50 nM of the 7-mer-A$_{20}$ RNA and 50 nM of the CCR4-NOT$_{MINI}$ complex. MBP-Strep alone or MBP-RNF219-C were titrated into the reaction (0.05–5.0 µM). Reactions were stopped after 32 min. Source data for panels (**b**–**f**) are provided as a Source Data file.

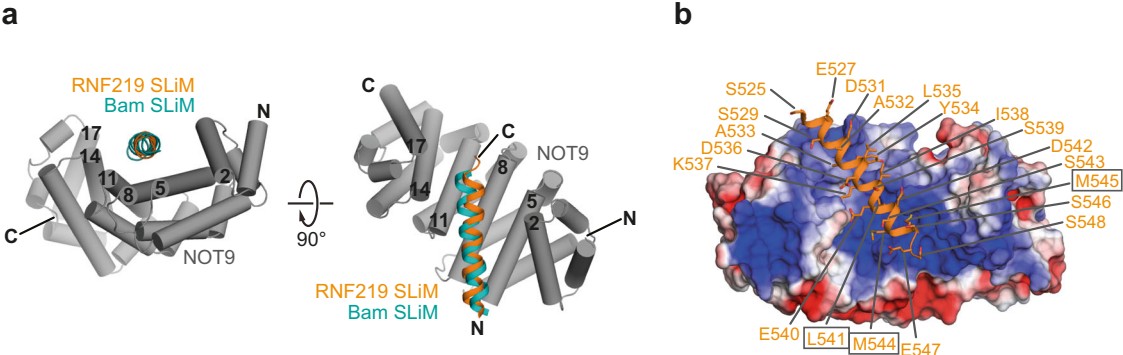

**Fig. 5 Structure prediction of the RNF219 SLiM in complex with NOT9. a** AlphaFold2-predicted structure of the RNF219 SLiM (orange cartoon) bound to NOT9 (gray) shown in two orientations. NOT9 helices are depicted as tubes and are numbered in black. The *Dm* Bam SLiM (cyan cartoon) is superposed on RNF219. **b** Electrostatic surface potential (with red representing negatively charged patches and blue positive ones) plotted onto the molecular surface of NOT9. The predicted bound RNF219 SLiM is shown as a cartoon (orange) with side chains of residues in stick representation. Visible residues are indicated with labels and residues substituted for glutamic acid are indicated in boxes.

RNF219 that are likely to engage the NOT9 pocket. To validate the predicted interface, we substituted the three residues with glutamic acid, and tested the SLiM-3E mutant in pull-down assays. Strikingly, the 3E substitution abolished binding of a minimal, MBP-tagged RNF219 fragment (residues 434-600) with the NOT9 module to the same extent as the deletion of the entire SLiM (Fig. 6a, b). To further validate the predicted RNF219-NOT9 interface, we compared the inhibitory profiles of the RNF219 434-600 region with the ΔSLiM and SLiM-3E counterparts. This revealed that the 3E substitution eliminated the ability of the RNF219 434-600 region to inhibit CCR4-NOT-mediated deadenylation in vitro to the same extent as deletion of the entire SLiM (Fig. 6c, d). Given that the SLiM residues predicted to contact NOT9 are fully conserved among vertebrates (Supplementary Fig. 12c), this motif may represent an evolutionarily conserved mode of NOT9 binding, which could not be previously identified by sequence analysis alone[41].

**RNF219 is a negative regulator of global mRNA degradation.** We then set out to investigate the role of RNF219 in cellular mRNA turnover using parental and RNF219 KO HeLa cells. In addition, we established rescue cell lines that stably express 3xFS-RNF219-WT or the catalytically inactive RING finger mutant RNF219-RINGmut in the KO background (Fig. 7a). The level of 3xFS-RNF219-WT was approximately nine-fold higher than that of endogenous RNF219, while the level of 3xFS-RNF219-RINGmut was approximately 14-fold higher (Fig. 7a), consistent with its enhanced stability and lack of autoubiquitination activity (Fig. 3b, Supplementary Fig. 6b). Importantly, both WT and RINGmut 3xFS-RNF219 were found to co-sediment with core CCR4-NOT subunits upon glycerol gradient centrifugation (Supplementary Fig. 13a, b), demonstrating that the ectopically expressed proteins are properly incorporated into the endogenous CCR4-NOT complex.

To determine transcriptome-wide mRNA half-lives by RNA-Seq, total RNA was extracted from the four cell lines at regular time intervals after transcriptional shut-off with actinomycin D (actD). Following depletion of ribosomal RNA and preparation of libraries for RNA-Seq, mRNA half-lives were calculated based on normalizing read counts to external RNA spike-in controls. By assuming first-order decay kinetics and requesting a coefficient of determination ($R^2$) > 0.5 for the analysis of log-transformed read counts by linear regression, reliable half-lives could be calculated for 1909 mRNAs from two independent biological replicates (Supplementary Data 2). The result of this analysis showed that the loss of RNF219 led to a global reduction in mRNA half-lives,

with most mRNAs below the diagonal (Fig. 7b). This corresponds to a reduction of the median mRNA half-life from 4.9 h in parental HeLa cells to 4.0 h in RNF219 KO cells. Interestingly, this global effect could be rescued by ectopic expression of both RNF219-WT and RNF219-RINGmut (Fig. 7c, Supplementary Fig. 14a, b), indicating that RNF219 acts as a negative regulator of global mRNA turnover in cells independently of its E3 ligase activity. These results are in line with our observation that RNF219-C, which lacks the N-terminal RING domain, exerts an inhibitory effect on the deadenylation activity of the CCR4-NOT complex in vitro (Fig. 4).

When the effect of RNF219 depletion was assessed for mRNAs grouped according to their half-lives, the destabilizing effect was observed for all categories (Fig. 7d), albeit long-lived mRNAs ($t_{1/2} > 12$ h) displayed a stronger dependency on RNF219 expression. Notably, RNF219-WT restored the stability of all mRNAs, whereas RNF219-RINGmut was able to restore the stability of short-lived mRNAs ($t_{1/2} \leq 6$ h), but not of the more long-lived mRNAs ($t_{1/2} > 9$ h). Since the RNF219 rescue pattern of short-lived mRNAs resembled that of global mRNA (Fig. 7c), we focused on more labile subgroups of mRNAs that are recruited to the CCR4-NOT complex by different RBPs. On the one hand, mRNAs associated with the m6A-binding protein YTHDF2[72] (Supplementary Fig. 14c, Supplementary Data 3), the AU-rich element-binding protein TTP[73] (Supplementary Fig. 14d, Supplementary Data 3) or the stem-loop-binding protein Roquin[74] (Supplementary Fig. 14e, Supplementary Data 3) were destabilized in RNF219 KO cells to a similar degree as bulk mRNA (Fig. 7c). On the other hand, mRNAs transcribed from immediate early genes (IEGs)[75], which are more labile than the other subgroups, showed stronger destabilization upon RNF219 KO (Fig. 7e, Supplementary Data 3). Interestingly, the half-lives of these mRNAs were substantially increased in cells overexpressing RNF219-RINGmut (Fig. 7e), an observation that we verified by measuring the half-lives of three IEG mRNAs, *KLF4*, *MYC* and *RHOB*, by qRT-PCR (Fig. 7f and Supplementary Fig. 14f). Taken together, these analyses led us to conclude that RNF219 acts as a general inhibitor of the CCR4-NOT deadenylase complex that attenuates the degradation of mRNAs transcriptome-wide, with a pronounced effect on IEG mRNAs.

Next, we asked whether RNF219 participates in regulating acetylation-induced mRNA turnover[53]. To this end, we compared parental HeLa cells with our stable HeLa cell line expressing elevated levels of RNF219-RINGmut (Fig. 7g). *GAPDH* and *NCL*, two very stable mRNAs under control conditions, showed accelerated decay with half-lives of approximately five hours

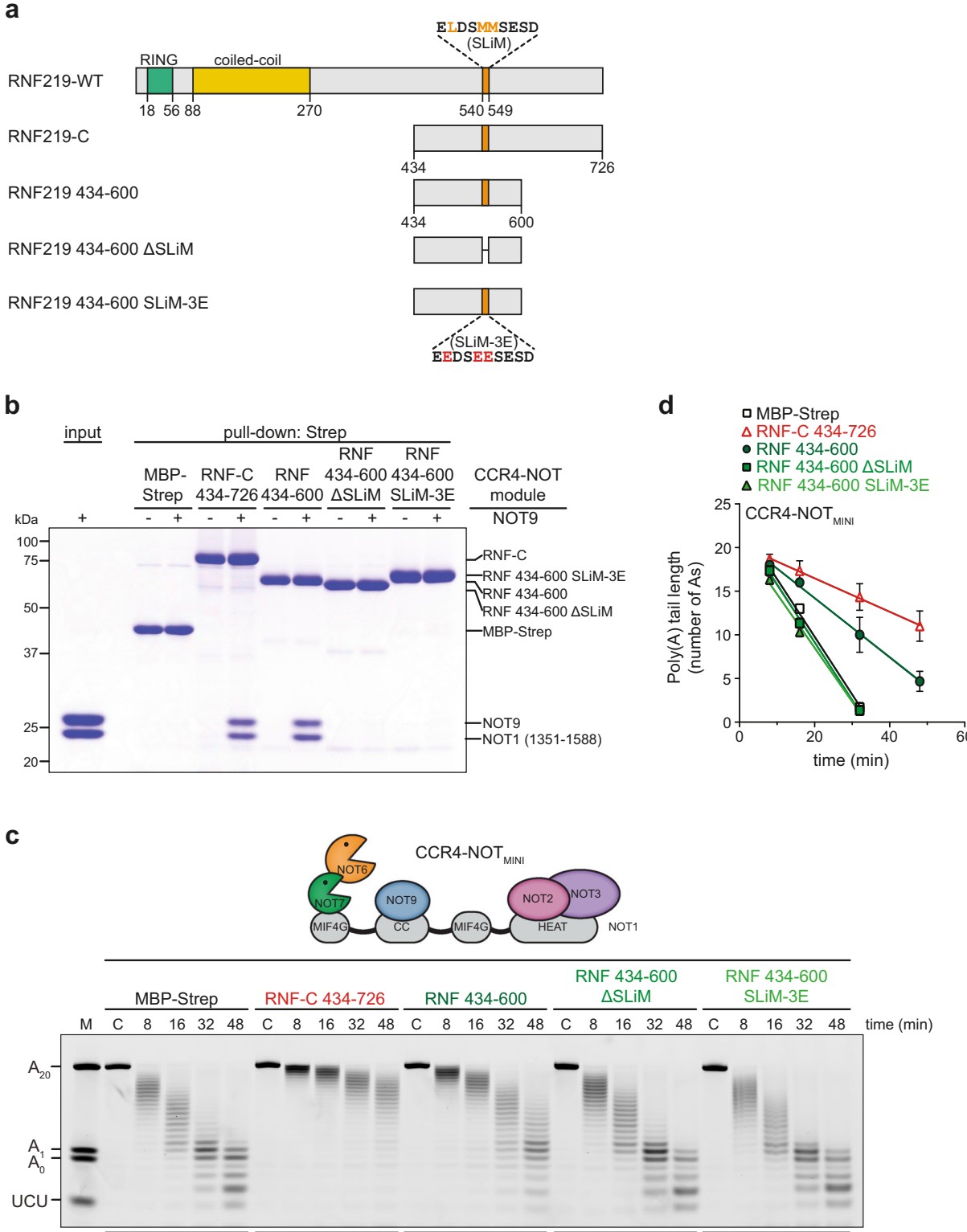

when parental HeLa cells were treated with the HDAC inhibitor RMD (Fig. 7h), akin to the destabilization observed by KD of HDAC1 and HDAC2[53]. In the RNF219-RINGmut expressing cells, however, *GAPDH* and *NCL* mRNA were less labile upon treatment with RMD (Fig. 7h and Supplementary Fig. 14g). Given that RMD treatment causes a pronounced drop in RNF219 expression (Fig. 1 and Supplementary Fig. 4a, b), these results

indicate that reduced association of RNF219 with the CCR4-NOT complex upon pharmacological HDAC inhibition contributes to acetylation-induced mRNA turnover.

Since accelerated mRNA turnover upon HDAC inhibition involves direct acetylation of the exoribonuclease CAF1a[53], we finally wanted to know whether RNF219-mediated inhibition of mRNA decay is dependent on the acetylation status of CAF1a. To

**Fig. 6 The SLiM is essential for the inhibitory function of RNF219. a** Schematic illustration of the domain architecture of RNF219; RNF219-C (residues 434-726), RNF219 434-600, RNF219 434-600 ΔSLiM and RNF219 434-600 SLiM-3E are shown below. Mutated amino acid residues in RNF219 434-600 SLiM-3E are highlighted in red. **b** Coomassie-stained polyacrylamide gel of in vitro pull-down assays with MBP-Strep alone or four recombinant RNF219 constructs fused to MBP and a StrepII (Strep) affinity tag after incubation with the NOT9 module. **c** In vitro deadenylation assay with 50 nM of a 7-mer-$A_{20}$ RNA substrate, 50 nM CCR4-NOT$_{MINI}$ complex and 2250 nM of MBP-Strep or indicated MBP-RNF219 constructs. Reactions were stopped at the indicated time points and analyzed by electrophoresis using a 20% denaturing polyacrylamide gel; M, RNA size marker; C, RNA alone control. **d** Quantification of the change in poly(A) tail length by plotting the most abundant tail length at each time point. Linear regression was used to determine the apparent deadenylation rate (As/min); values are presented as mean ± SE ($n = 3$). Source data for panels (**b**–**d**) are provided as a Source Data file.

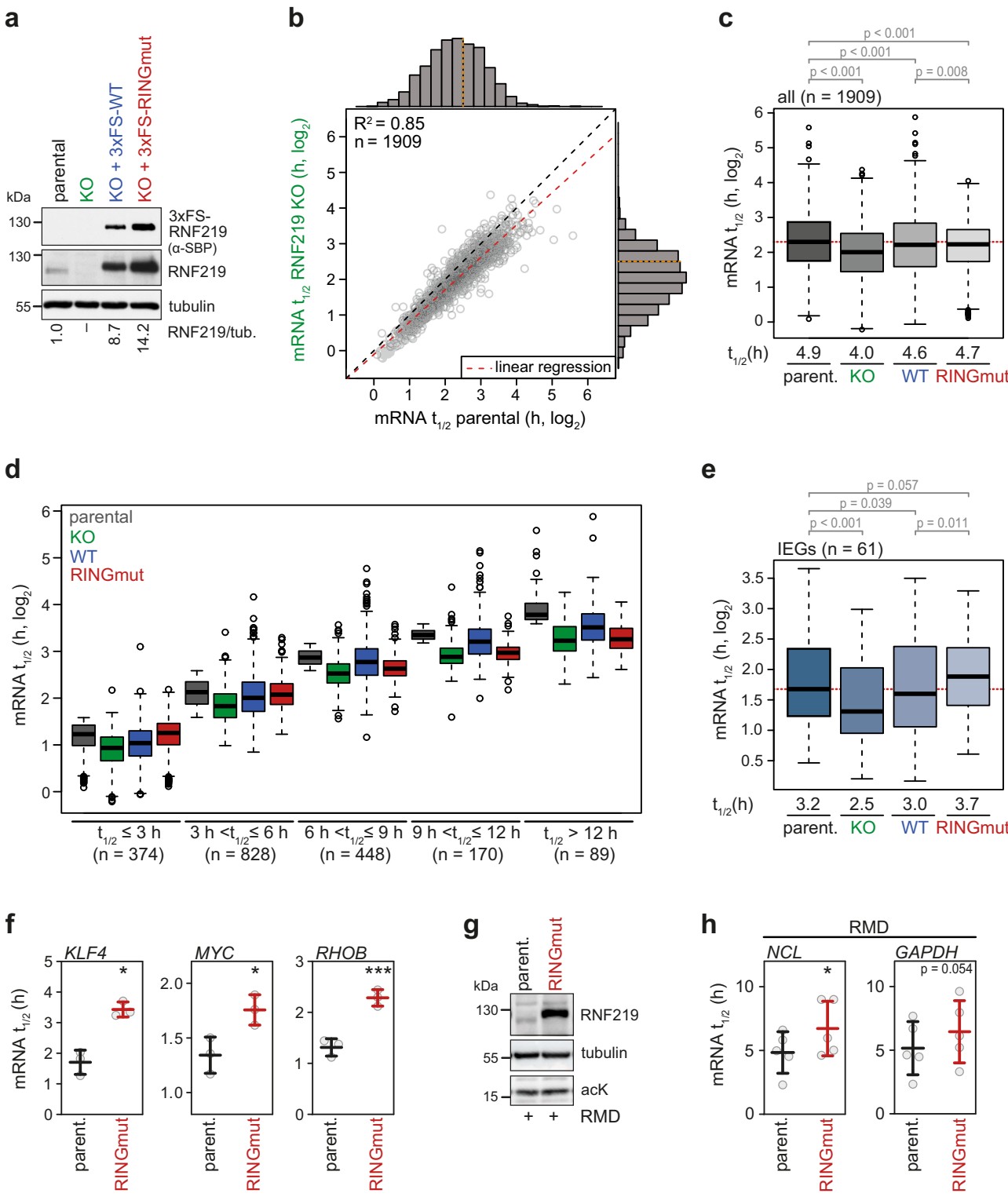

**Fig. 7 RNF219 attenuates global mRNA decay. a** Western blot analysis of RNF219 expression in parental HeLa, RNF219 KO cells and RNF219 KO cells stably expressing 3xFS-RNF219-WT or -RINGmut using antibodies as indicated. The relative RNF219 expression level is given below; tubulin serves as loading control. **b** Scatter plot and marginal histograms depicting mRNA half-lives in RNF219 KO vs. parental HeLa cells. Transcriptome-wide mRNA stability measurements were performed following transcription shut-off with 5 µg/ml actinomycin D (actD) and RNA-Seq analysis of rRNA-depleted total RNA extracted at regular time intervals ($n = 2$). **c** Box-and-whisker plot depicting half-lives of all mRNAs determined in (**b**). The dotted line shows the median mRNA half-life in parental HeLa cells; $p$ values were calculated by two-sided, paired Wilcoxon rank sum test; parent. vs. KO $p = 1.2 \times 10^{-240}$; parent. vs. WT $p = 4.1 \times 10^{-28}$; parent. vs. RINGmut $p = 1.5 \times 10^{-80}$; WT vs. RINGmut $p = 0.008$. **d** Box-and-whisker plot depicting half-lives of mRNAs from (**c**) binned into discrete intervals as indicated. **e** Box-and-whisker plot depicting half-lives of IEG mRNAs as in (**c**); $p$ values were calculated by two-sided, paired Wilcoxon rank sum test; parent. vs. KO $p = 1.3 \times 10^{-10}$; parent. vs. WT $p = 0.039$; parent. vs. RINGmut $p = 0.057$; WT vs. RINGmut $p = 0.011$. **f** Half-life measurement of IEG mRNAs *KLF4*, *MYC* and *RHOB* in parental HeLa cells and RNF219 KO cells stably expressing 3xFS-RNF219-RINGmut. Total RNA was prepared as described in (**b**) and analyzed by qRT-PCR; values are presented as mean ± SD ($n = 3$; KLF4 $p = 0.0139$; MYC $p = 0.037$; RHOB $p = 0.0007$). **g** Western blot analysis of RNF219 expression in RNF219 KO cells and RNF219 KO cells stably expressing 3xFS-RNF219-RINGmut following treatment with 20 nM RMD for 16 h. **h** Half-life measurement of normally stable *GAPDH* and *NCL* mRNA in parental HeLa cells and RNF219 KO cells stably expressing 3xFS-RNF219-RINGmut following treatment with 20 nM RMD for 16 h, as in (**f**). Values are presented as mean ± SD ($n = 5$; NCL $p = 0.0405$; GAPDH $p = 0.0543$). $p$ values in (**f**) and (**h**) were calculated by two-tailed, paired *t* test (*$p < 0.05$; ***$p < 0.001$). In all box-and-whisker plots the median is indicated as center line, the interquartile range as box and the whiskers extend to the most extreme datapoint that is no >1.5 times the interquartile range from the box. Source data for panels (**a–c**) and (**e–h**) are provided as a Source Data file.

---

test this, we co-expressed 3xFS-RNF219-RINGmut together with CAF1a-K$_4$A (Supplementary Fig. 15a), a hypoactive CAF1a mutant with reduced levels of acetylation[53], and measured Roquin-induced degradation of a constitutive decay element (CDE)[2]-containing *β-globin* reporter mRNA. The analysis revealed that expression of 3xFS-RNF219-RINGmut slows down reporter mRNA decay in the presence of CAF1a-WT (Supplementary Fig. 15b–d; half-life of 9.9 ± 2.3 vs. 15.7 ± 3.0 h). Importantly, 3xFS-RNF219-RINGmut also slows down reporter mRNA decay in the presence of CAF1a-K$_4$A (half-life of >24 vs. >50 h), indicating that RNF219-mediated inhibition of mRNA deadenylation and decay is independent of CAF1a acetylation.

In conclusion, our mRNA decay analyses provide compelling evidence that RNF219 functions as an inhibitor of global mRNA turnover, whose reduced expression upon HDAC inhibition contributes to acetylation-induced mRNA turnover independently of CAF1a acetylation.

## Discussion

The CCR4-NOT complex is a key regulator of post-transcriptional gene expression, responsible for both bulk and transcript-specific deadenylation of mRNAs during co- and post-transcriptional mRNA decay[76–78]. To fulfill these tasks, the CCR4-NOT complex is embedded within a protein interaction network comprising a multitude of mRNA decay-promoting RBPs that provide target specificity[3,51,79]. In addition, the CCR4-NOT complex is connected to general activators of deadenylation such as proteins of the BTG/TOB family, which recruit mRNAs by associating with the cytoplasmic poly(A)-binding protein PABPC1[1,50,54,80]. In contrast, less is known about regulatory factors that directly modulate the activity of the CCR4-NOT deadenylase. The DEAD-box helicase DDX6, which serves as an activator for mRNA decapping and is important for miRNA-mediated silencing of mRNAs, directly binds to NOT1[32,33,81] and was recently found to enhance the deadenylating activity of CAF1[82]. The same study showed that another DEAD-box helicase, eIF4A2, also binds to NOT1, but inhibits CAF1-mediated deadenylation and maintains long poly(A) tails on bound mRNAs.

During the course of our study, two laboratories provided independent evidence that RNF219 associates with the human CCR4-NOT complex and negatively regulates deadenylation. Guénolé et al. (2019) showed that tethering of RNF219 to a reporter mRNA represses its translation and causes elongation of the poly(A) tail. Du et al. (2020) showed that overexpression of RNF219 reduces the deadenylation rate of reporter mRNAs

containing a miRNA-binding site or an m$^6$A target site and identified a role for RNF219 in controlling gene expression during neuronal differentiation. Moreover, RNF219 was identified as a binding partner of the CCR4-NOT complex when using the CAF1-interacting protein TOB2 as a bait[54]. Our work presented here establishes RNF219 as an acetylation-regulated cofactor of the CCR4-NOT deadenylase complex (Fig. 1). We demonstrate that RNF219 directly and stably binds to the complex via the NOT9 module, and this interaction depends on a conserved SLiM within the C-terminal low-complexity region of RNF219 (Figs. 2 and 6 and Supplementary Fig. 12c). Moreover, our in vitro deadenylation assays provide firm evidence that RNF219 is an antagonist of a reconstituted six-subunit human CCR4-NOT complex comprising the scaffold subunit NOT1, the NOT6/NOT7 catalytic module, the central NOT9 module and the C-terminal NOT module (Fig. 4). Structure prediction of the RNF219 SLiM revealed that it adopts the fold of an amphipathic α-helix, which engages the concave and positively charged surface of NOT9 (Fig. 5). Remarkably, structure-guided mutagenesis confirmed that RNF219-mediated inhibition of deadenylation depends on the direct interaction of RNF219 with NOT9, with the crucial three hydrophobic amino acids located in the C-terminal half of the SLiM (Fig. 6).

Using transcriptome-wide mRNA half-life measurements, our study reveals for the first time that RNF219 indeed acts as an inhibitor of global mRNA decay *in cellulo* (Fig. 7). Given that both RNF219-WT and RNF219-RINGmut were able to rescue the global defect in mRNA stability (Fig. 7c and Supplementary Fig. 14a, b), RNF219 does not require its E3 ligase activity for acting as an inhibitor of the CCR4-NOT deadenylase on bulk mRNA. This agrees with the observation that RNF219-mediated inhibition of deadenylation in vitro does not require the N-terminal RING domain (Fig. 4). However, our results also showed that the integrity of the RNF219 RING domain is needed for stabilization of long-lived mRNAs in cells (Fig. 7d), indicating that the E3 ligase activity may influence subgroups of mRNAs beyond its role in destabilizing the RNF219 protein itself through autoubiquitination (Fig. 3b).

Interestingly, the group of short-lived IEG mRNAs was stabilized to a greater extent by RNF219-RINGmut than by RNF219-WT (Fig. 7e). Not only does this show that the RING domain is not required for stabilizing IEG mRNAs, but it further indicates that the abundance of RNF219 is important for antagonizing the degradation of these mRNAs. A possible scenario is that RNF219 competes for NOT9 binding with RBPs that serve as destabilizing adaptors of IEG mRNAs.

Downregulation of RNF219 expression by the HDAC inhibitor RMD reduces its incorporation into the CCR4-NOT complex (Fig. 1c–e, Supplementary Fig. 4), which is in good agreement with our earlier discovery that pharmacological HDAC inhibition leads to acceleration of poly(A) RNA degradation[53]. Indeed, we could show that elevated expression of RNF219-RINGmut antagonizes RMD-induced degradation of *NCL* and *GAPDH* mRNAs (Fig. 7h). From these results we conclude that loss of RNF219 expression contributes to the acetylation-induced degradation of normally stable mRNAs. Since RNF219 acts independently of CAF1 acetylation (Supplementary Fig. 15), its dissociation from the complex synergizes with the stimulatory effect that direct acetylation of CAF1 has on CCR4-NOT-mediated mRNA degradation[53].

Notably, our study revealed that RNF219 associates with the CCR4-NOT complex through a direct interaction with the NOT9 module (Fig. 2c). NOT9/Caf40 is a highly conserved six armadillo-repeat-containing protein[83], which serves as docking site for multiple CCR4-NOT-interacting RBPs such as TTP, GW182/TNRC6, NOT4, *Dm* Bam and *Dm* Roquin[32,33,37,39–41]. Our pull-down assays with recombinant proteins demonstrate that RNF219 interacts exclusively with the NOT9 module (Fig. 2c), unlike *Dm* Roquin, human NOT4 or human TTP, all of which engage through additional interactions with other subunits of the complex[37,40,84]. In line with our in vitro binding results, partial KD of NOT9 was sufficient to abolish the co-precipitation of RNF219 with the CCR4-NOT complex (Fig. 2b). Similar to *Dm* Bam[41] and Roquin[40], the RNF219-NOT9 interaction depends on a SLiM located in the C-terminal low-complexity region of RNF219 (Figs. 2d and 6b)[63]. This peptide motif contains a short stretch of highly conserved amino acids (Supplementary Fig. 12c) and occupies the same NOT9 binding surface as the SLiMs of *Dm* Bam[41] and Roquin[40] (Fig. 5a, Supplementary Fig. 12b). Hence, the RNF219-NOT9 interaction represents a structurally conserved mode of CCR4-NOT binding likely as a consequence of convergent evolution.

In vitro experiments using reconstituted human CCR4-NOT complexes have shown that NOT9, which is positioned proximal to the exonuclease module, acts as an RNA-interaction surface that has the potential to stimulate deadenylation through improved substrate binding[25]. One possibility is that RNF219 inhibits deadenylation by masking the solvent-exposed RNA-binding surface of NOT9, thereby displacing mRNAs non-specifically (Fig. 8). This model would help to explain why labile mRNAs, which rely on rapid and processive deadenylation, are

stabilized by RNF219 overexpression in this (Fig. 7e, f) and other studies[63,85]. Since the rescue of long-lived mRNAs was found to depend on the integrity of the RNF219 RING domain (Fig. 7d), an alternative possibility is that RNF219 may directly affect CAF1 in a context-specific manner, given the proximity of NOT9 to the catalytic module. Further experiments will be required to precisely determine how RNF219 attenuates CCR4-NOT dead-enylase activity, and why the RING domain is required for stabilizing long-lived, but not short-lived mRNAs.

Unlike yeast NOT4, which is an integral component of the CCR4-NOT complex and fulfills versatile functions including ubiquitin-mediated proteolysis, co-translational quality control and proteasome assembly[78,86], human and *Dm* NOT4 do not co-purify with the CCR4-NOT complex (Fig. 1b)[24,38]. In contrast to GW182/TNRC6 and TTP, which interact with tryptophan binding pockets located on the convex surface of NOT9[32,33,39], NOT4 utilizes a C-terminal SLiM, the so called Caf40-binding motif, to associate with the concave surface of NOT9[37]. Since RNF219 also interacts with the concave surface of NOT9 via a SLiM within its C-terminal low-complexity region (Figs. 2d, 5 and 6), NOT4 might compete with RNF219 for the same binding site as it has been suggested for *Dm* Bam and *Dm* Roquin[37]. Since RNF219 is tightly associated with the human CCR4-NOT complex (Fig. 1b, Supplementary Fig. 3c), yet its RING domain is only required for attenuating the degradation of long-lived transcripts (Fig. 7d), it will be interesting to explore whether ubiquitination associated with the mammalian CCR4-NOT complex might regulate the stability of distinct mRNA subsets. Based on the example of the RNA-binding E3 ligase MEX-3C, which was found to facilitate the deadenylation of MHC-I mRNA by ubiquitination of CAF1a/NOT7[52], one is tempted to speculate that the E3 ligase activity of RNF219 may control the deadenylation of specific mRNAs beyond the RING domain-independent role of RNF219 in attenuating global mRNA decay.

## Methods

**Cell culture.** HeLa and HEK293 cells (a kind gift from Paul Anderson, Harvard Medical School) were cultured in Dulbecco's Modified Eagle Medium (DMEM) supplemented with 10% (v/v) fetal bovine serum (Sigma-Aldrich), 2 mM L-glutamine, 100 U/ml penicillin and 0.1 mg/ml streptomycin (all PAN Biotech) and incubated at 37 °C in a 5% CO$_2$ incubator. Cell lines were authenticated via SNP profiling by Multiplexion GmbH at DKFZ. Cell lines were regularly tested for mycoplasma contamination using a PCR Mycoplasma Test Kit (AppliChem).

*Spodoptera frugiperda* Sf21 insect cells (a kind gift from Imre Berger, University of Bristol) were cultured in Sf900II medium (Thermo Fisher Scientific) and incubated at 27 °C on a shaking platform.

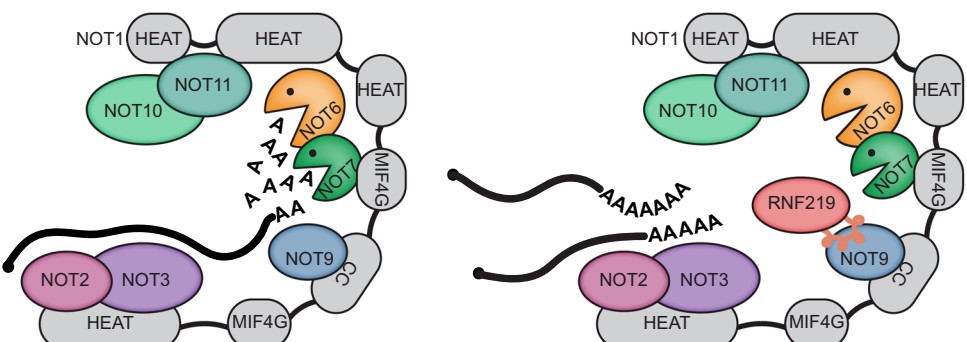

**Fig. 8 Model of CCR4-NOT-mediated deadenylation and its inhibition by RNF219.** The non-catalytic NOT2/3 and NOT9 modules facilitate access of substrate poly(A) RNA to the catalytic NOT6/NOT7 module and thereby promote sequence-independent mRNA deadenylation. RNF219 interacts with the NOT9 module via a SLiM (depicted as a helical peptide) within its C-terminal low-complexity region. RNF219 may occupy the RNA interaction surface on NOT9 and hinder access of poly(A) RNA to the catalytic module. Alternatively, RNF219 may reduce the catalytic activity of CCR4 or CAF1, given the proximity of the NOT9 and the catalytic module. Inhibition of CCR4-NOT-mediated deadenylation by RNF219 occurs in the absence of its N-terminal RING domain and thus, is independent of its E3 ligase activity. The N-terminal RING domain of RNF219 is dispensable for stabilization of short-lived mRNAs yet required for stabilization of long-lived mRNAs.

**Reverse siRNA transfection.** Knock-down experiments were performed with Lipofectamine RNAiMAX transfection reagent (Thermo Fisher Scientific) and reverse transfection. siRNAs were used at a final concentration of 100 nM and diluted in Opti-MEM reduced serum medium (Thermo Fisher Scientific). For NOT9 knock-down, cells were transfected with siRNA over a period of 48 h. For NOT1 and HDAC1 knock-down, cells were transfected twice with siRNA over a period of 72 h. siRNAs were designed with help of the siDESIGN Center (Horizon Discovery) and synthesized by Eurofins Genomics. All siRNA sequences are listed in Supplementary Data 4.

**Plasmid transfection.** HeLa cells were seeded in 10 cm dishes at a density of $1.5 – 1.8 \times 10^6$ cells/dish and transfected the following day with plasmid DNA diluted in Opti-MEM reduced serum medium at a ratio of 1:3 using Lipofectamine 3000 transfection reagent (all Thermo Fisher Scientific). The medium was exchanged 4 h post transfection. HEK293 cells were seeded in 15 cm dishes at a density of $5 \times 10^6$ cells/dish and transfected as above.

**CRISPR/Cas9-mediated genome editing.** Target specific guide RNA sequences were designed with help of CRISPR design tools (https://zlab.bio/guide-design-resources) and are listed in Supplementary Data 4.

For tagging of endogenous NOT1, HeLa cells were transfected with an equal amount of Cas9-guide RNA expressing constructs pSpCas9(BB)-2A-GFP-*CNOT1*-guideC (p3568) and pSpCas9(BB)-2A-GFP-*CNOT1*-guideB (p3544) together with the 3xFLAG-2xStrep(FST)-*CNOT1* repair template containing mutated PAMs (p3598) at a ratio of 4:1 using Lipofectamine 3000 transfection reagent (Thermo Fisher Scientific). Cells were treated with 10 μM RS-1 (Sigma–Aldrich) for 24 h to enhance CRISPR/Cas9-mediated genome editing efficiency. Two days after transfection, GFP$^+$ cells were enriched by FACS (FlowCore, Medical Faculty Mannheim). Single clones were obtained by limiting dilution cloning using 96-well plates and 10% (v/v) conditioned medium. Integration of the FST tag was monitored in single clones by western blot analysis and sequencing of the genomic locus.

To generate RNF219 KO cells, HeLa cells were transfected with an equal amount of Cas9-guide RNA expressing constructs pSpCas9(BB)-2A-GFP-*RNF219*-guideA (p3719) and pSpCas9(BB)-2A-GFP-*RNF219*-guideC (p3721) using Lipofectamine 3000 transfection reagent and processed as described above. Successful CRISPR/Cas9-mediated genome editing was monitored in single clones by PCR and sequencing of the genomic locus.

NOT9 KO cells were generated by nucleofection of Alt-R CRISPR/Cas9 ribonucleoprotein complexes (Integrated DNA Technologies). Briefly, each crRNA was resuspended at a concentration of 200 μM in 1x TE and mixed with 200 μM tracrRNA (IDT) at an equimolar ratio. RNA hybridization was achieved by incubation at 95 °C for 5 min, followed by stepwise cooling to room temperature. CRISPR/Cas9 RNP complexes were prepared by gently mixing hybridized RNAs with Cas9 protein (IDT) at a 1.2:1 molar ratio followed by 20 min incubation at room temperature. For each nucleofection reaction, $1 \times 10^6$ Hela cells were washed once in PBS and then resuspended in 90 μl supplemented solution R (Lonza) followed by the addition of 4 μl electroporation enhancer (IDT) and 2.5 μl of each prepared RNP complex. Nucleofection reactions were transferred into 100 μl cuvettes (Lonza) and pulsed in an Amaxa 2b nucleofector using program A-28. Following 48 h of incubation in conditioned medium, single clones were seeded into 96-well plates as described above. Successful CRISPR/Cas9-mediated genome editing was monitored in single clones by western blotting and sequencing of the genomic locus.

**Lentiviral transduction.** To generate RNF219 KO cells stably expressing 3xFS-RNF219-WT, -RINGmut or -ΔSLiM, lentiviral particles were produced in HEK293 cells by co-transfection of lentiviral expression vectors pWPI-3xFLAG-SBP-RNF219-WT (p3761), -RINGmut (p3762) or -ΔSLiM (p3802) together with the lentiviral packaging vector pCMV-Δ8.91 (p3785) and the VSV-G expression vector pMD2.G (p3786) at a ratio of 3:3:1 with TurboFectin 8.0 (OriGene). Growth medium was exchanged 6 h post transfection. After 2 days, virus-containing supernatant was filtrated through a 0.45 μm syringe filter and used for two consecutive rounds of transduction using HeLa RNF219 KO clone #16 as target cell line. Selection of stably transduced cells was performed with 6 μg/ml blasticidin S (AppliChem) for 7–14 days after the second transduction.

**Purification of the FST-tagged endogenous CCR4-NOT complex.** $2.2 \times 10^6$ parental or FST-NOT1 (clone #47) HeLa cells were seeded in 15 cm dishes. 36 h later, cells were washed once with ice-cold phosphate-buffered saline (PBS), harvested by scraping and flash-frozen in liquid nitrogen. For each condition, $10 \times 15$ cm dishes were used. Cells were mechanically disrupted using a Tissuelyzer II (4 × 15 sec, 25 Hz) with 5 mm stainless steel beads (both from Qiagen). Cell powder was solubilized in 9 ml ice-cold lysis buffer (20 mM Tris-HCl pH 7.5, 150 mM NaCl, 1.5 mM MgCl$_2$, 1 mM DTT, 0.05% [v/v] NP-40) supplemented with cOmplete EDTA-free Protease Inhibitor Cocktail (Roche). Lysates were vortexed briefly and incubated on a rotating shaker for 10 min at 4 °C. Lysates were then homogenized with 5 up and down strokes through a 27-gauge needle and cleared by centrifugation at 13,000 g for 20 min at 4 °C. 200 μl equilibrated anti-FLAG M2

magnetic beads (Sigma–Aldrich) were then added for 3 h at 4 °C on a rotating shaker, and beads were subsequently washed once with ice-cold lysis buffer and six times with ice-cold wash buffer (20 mM Tris-HCl pH 7.5, 150 mM NaCl, 2.5 mM MgCl$_2$, 1 mM DTT, 0.1% [v/v] NP-40). Bound proteins were eluted in 100 μl lysis buffer supplemented with 250 μg/ml 3xFLAG peptide (Sigma–Aldrich) for 1 h at 4 °C with vigorous shaking.

**Quantitative mass spectrometry of the CCR4-NOT complex.** FST-NOT1 (clone #47) HeLa cells were subjected to SILAC by differential labeling using the SILAC DMEM Lysine(8) Arginine(10) Kit (Silantes) and treated with 20 nM Romidepsin (Abcam) or an equal volume of solvent (DMSO) for 16 h. Cells were washed once with ice-cold PBS, harvested by scraping in ice-cold PBS supplemented with HDAC inhibitors (1 μM Trichostatin A [Cell Signaling Technology], 10 mM nicotinamide, 5 mM sodium butyrate [both from Sigma–Aldrich]), and subsequently flash-frozen in liquid nitrogen. Cell lysates were prepared as described above. 25–30 mg of total protein from each SILAC label were combined prior to affinity purification of FST-NOT1 with equilibrated anti-FLAG M2 magnetic beads as described above. Eluted proteins were mixed with 5x SDS sample buffer (250 mM HEPES pH 7.4, 10% [w/v] SDS, 50% [v/v] glycerol, 500 mM DTT, bromophenol blue) and incubated for 10 min at 95 °C. Purified CCR4-NOT complexes were shortly run into a 10% NuPAGE polyacrylamide Bis-Tris gel using NuPAGE MOPS SDS Running Buffer (both from Thermo Fisher Scientific) and stained with colloidal Coomassie (VWR). The gel was subjected to MS analysis at the ZMBH Mass Spectrometry and Proteomics Core Facility. The Coomassie-stained bands were cut out and processed as described with minor modifications[87]. In brief, trypsin digestion was done overnight at 37 °C. The reaction was quenched by addition of 20 μL of 0.1% trifluoroacetic acid (TFA; Biosolve, Valkenswaard, The Netherlands) and the supernatant was dried in a vacuum concentrator before LC-MS analysis. Nanoflow LC-MS2 analysis was performed with an Ultimate 3000 liquid chromatography system coupled to an QExactive HF mass spectrometer (Thermo Fisher Scientific). Samples were dissolved in 0.1% TFA, injected to a self-packed analytical column (75 μm x 200 mm; ReproSil Pur 120 C18-AQ; Dr Maisch GmbH) and eluted with a flow rate of 300 nl/min in an acetonitrile-gradient (3–40% [v/v]). The mass spectrometer was operated in data-dependent acquisition mode, automatically switching between MS and MS2. Collision induced dissociation MS2 spectra were generated for up to 20 precursors with normalized collision energy of 29%. Raw files were processed using MaxQuant v1.6.12.0[88] for peptide identification and quantification. MS2 spectra were searched against the Uniprot human proteome database (UP000005640_9606.fasta downloaded Nov 2019) and the contaminants database provided together with the MaxQuant software using the Andromeda search engine with the following parameters: Carbamidomethylation of cysteine residues as fixed modification and acetyl (protein N-term), oxidation (M) and deamidation (Q, N) as variable modifications, trypsin/P as the proteolytic enzyme with up to 2 missed cleavages was allowed. The maximum false discovery rate for proteins and peptides was 0.01 and a minimum peptide length of 7 amino acids was required. All other parameters were default parameters of MaxQuant.

**Analysis of mass spectrometry data.** Downstream data analysis was performed using LFQ values in Perseus v1.6.2.1[89]. Commonly occurring contaminants, proteins only identified by a modification site or those matching the reversed part of the decoy database were excluded. To exclude proteins identified with low confidence, proteins were filtered for LFQ values being present in at least 3 samples across four biological repeat experiments. Means of log$_2$-transformed LFQ values were calculated and core CCR4-NOT subunits as well as candidate interactors were revealed based on their exclusive identification in FST-NOT1 HeLa cells. SILAC ratios were used to identify acetylation-regulated interactors of the CCR4-NOT complex.

**Glycerol gradient fractionation.** HeLa cells were harvested and mechanically lysed as described above. Cell powder was solubilized in ice-cold lysis buffer (10 mM Tris-HCl pH 7.5, 10 mM NaCl, 10 mM MgCl$_2$, 0.1% [v/v] Igepal CA-630) supplemented with cOmplete EDTA-free Protease Inhibitor Cocktail (Roche). Lysates were vortexed briefly and incubated on a rotating shaker for 10 min at 4 °C. Lysates were then homogenized with 5 up and down strokes through a 27-gauge needle and cleared by centrifugation at 100,000 g for 1 h at 4 °C. 3–4 mg total protein was loaded onto linear gradients of 10–30% [w/v] glycerol dissolved in 20 mM Tris-HCl pH 8.0, 200 mM K(OAc), 5 mM Mg(OAc)$_2$, 1 mM DTT supplemented with cOmplete EDTA-free Protease Inhibitor Cocktail. Samples were centrifuged at 229,900 g for 18 h at 4 °C using a swinging bucket rotor (Hitachi). Fractions were eluted from the top of the gradient using a density gradient fractionation system (Brandel). During elution, fractions of ~900 μl were collected. Fractions were then supplemented with 300 μl 20 mM Tris-HCl pH 7.5 and 15 μl StrataClean Resin (Agilent) and incubated overnight on a rotating shaker at 4 °C. Proteins were eluted from the resin with 2x SDS sample buffer (100 mM HEPES pH 7.4, 4% [w/v] SDS, 20% [v/v] glycerol, 200 mM DTT, bromophenol blue).

**Co-immunoprecipitation.** Cells were harvested by scraping in ice-cold PBS, centrifuged at 400 g for 3 min at 4 °C and immediately flash-frozen in liquid nitrogen.

Following cryogenic lysis with a Tissuelyzer II (4 × 15 sec, 25 Hz), cell powder was solubilized in co-IP lysis buffer (20 mM Tris-HCl pH 7.5, 150 mM NaCl, 1.5 mM MgCl₂, 1 mM DTT, 0.05% [v/v] NP-40) supplemented with cOmplete EDTA-free Protease Inhibitor Cocktail (Roche) and vortexed. Following incubation on a rotating shaker for 10 min at 4 °C, lysates were cleared by centrifugation at 425 g for 5 min at 4 °C. 1% of total lysate was saved as input. Equal amounts of total protein were used for IP using anti-FLAG M2 magnetic beads (Sigma–Aldrich), Dynabeads MyOne Streptavidin C1 or anti-HA magnetic beads (both Thermo Fisher Scientific) for 3 h at 4 °C on a rotating shaker. Beads were washed once with ice-cold lysis buffer and six times with ice-cold wash buffer (20 mM Tris-HCl pH 7.5, 150 mM NaCl, 2.5 mM MgCl₂, 1 mM DTT, 0.1% [v/v] NP-40). Bound proteins were eluted with 2x SDS sample buffer.

**Purification of 3xFS-RNF219 from HEK293 cells.** HEK293 cells were seeded in 15 cm dishes at a density of $5 \times 10^6$ cells/dish and transfected the following day with 5 μg of plasmid DNA diluted in Opti-MEM reduced serum medium at a ratio of 1:3 using Lipofectamine 3000 transfection reagent (all Thermo Fisher Scientific). The medium was changed 4 h post transfection. Cells were harvested and lysed in co-IP lysis buffer as described above. The lysis buffer was additionally supplemented with 100 μM ZnCl₂. RNF219 was immunoprecipitated with anti-FLAG M2 magnetic beads (Sigma–Aldrich) for 2–3 h at 4 °C. Beads were washed once with ice-cold-lysis buffer and six times with wash buffer (20 mM Tris-HCl pH 7.5, 300 mM NaCl, 2.5 mM MgCl₂, 1 mM DTT, 0.2% [v/v] NP-40, 20 μM ZnCl₂). RNF219 was eluted with 250 μg/ml 3xFLAG peptide (Sigma–Aldrich) in in vitro ubiquitination buffer (30 mM Tris-HCl pH 7.5, 100 mM NaCl, 1 mM DTT, 5 mM MgCl₂) for 1 h at 4 °C with vigorous shaking.

**Purification of 3xFS-RNF219 from NOT9 KO cells.** $2.5 \times 10^6$ CRISPR control (clone #4) or NOT9 KO (clone #5) HeLa cells stably expressing 3xFS-RNF219 were seeded in 15 cm dishes. 36 h later, cells were washed once with ice-cold PBS, harvested by scraping and flash-frozen in liquid nitrogen. Cells were harvested and lysed in co-IP lysis buffer as described above. Equal amounts of total protein were used for the purification of 3xFS-RNF219 using anti-FLAG M2 magnetic beads (Sigma–Aldrich) for 3 h at 4 °C on a rotating shaker. Beads were washed once with ice-cold-lysis buffer and six times with wash buffer (20 mM Tris-HCl pH 7.5, 150 mM NaCl, 2.5 mM MgCl₂, 1 mM DTT, 0.2% [v/v] NP-40). RNF219 was eluted with 250 μg/ml 3xFLAG peptide (Sigma–Aldrich) in in vitro ubiquitination buffer as described above.

**Western blot analysis.** For protein separation, 5–20% polyacrylamide gradient Tris-glycine gels were made using standard protocols. The samples to be analyzed were mixed with an appropriate volume of 5x or 2x SDS Laemmli buffer and denatured for 10 min at 95 °C. 15–30 μg of total protein was loaded per sample and electrophoresis was performed in Tris-glycine SDS running buffer (25 mM Tris, 250 mM glycine, 0.1% [w/v] SDS) at 30 mA/gel. Proteins were transferred onto 0.2 μm pore-sized nitrocellulose membranes (Peqlab) in Tris-glycine blotting buffer (20 mM Tris, 150 mM glycine, 20% EtOH) at 90 V for 3 h at 4 °C by using a wet blotting device. Loading and blotting efficiency were monitored by Ponceau S staining of membranes. Following de-staining with TBS-T (50 mM Tris-HCl pH 7.5, 150 mM NaCl, 0.1% [v/v] Tween-20), membranes were blocked with 5% [w/v] milk/PBS/0.01% [w/v] sodium azide for 1 h at room temperature on a shaking platform. Membranes were incubated with primary antibodies diluted in PBS/ 0.01% [w/v] sodium azide overnight at 4 °C on a shaking platform. The following primary antibodies were used: mouse monoclonal anti-FLAG M2 (1:1000, Sigma–Aldrich), rabbit polyclonal anti-histone H3 acetyl K27 (1:1000, Abcam), rabbit polyclonal anti-NOT10 (1:1000, Proteintech), rabbit monoclonal anti-NOT7 (1:1000, Cell Signaling), rabbit polyclonal anti-acetylated lysine (1:1000, Cell Signaling), rabbit polyclonal anti-RNF219 (1:500, Bethyl), rabbit polyclonal anti-FHL2 (1:1000, Abcam), rabbit polyclonal anti-TNKS1BP1 (1:1000, Bethyl), rabbit monoclonal anti-NOT2 (1:1000, Cell Signaling), rabbit polyclonal anti-NOT9 (1:1000, Proteintech), mouse monoclonal anti-SBP (1:1000, Santa Cruz), rat monoclonal anti-tubulin (1:1000, Abcam), rabbit monoclonal anti-NOT6 (1:1000, Cell Signaling), rabbit polyclonal anti-histone H3 (1:1000, Abcam), rabbit polyclonal anti-NOT1 (1:1000, Proteintech), rabbit polyclonal anti-CAF1a (1:1000, kindly provided by Ann-Bin Shyu, McGovern Medical School, University of Texas, Houston), rabbit monoclonal anti-NOT3 (1:1000, Cell Signaling), mouse monoclonal anti-ubiquitin (1:1000, Cell Signaling), rabbit monoclonal anti-CAPZA1 (1:1000, Abcam), rabbit monoclonal anti-CAPZB (1:1000, Abcam), rabbit polyclonal anti-GFP (1:1000, Abcam), mouse monoclonal anti-HDAC1 (1:1000, Santa Cruz). Membranes were washed 5–6 times over a time period of 1 h with TBS-T before incubation with horseradish peroxidase (HRP)-coupled secondary antibodies (1:10,000, donkey anti-rabbit, donkey anti-mouse, donkey anti-rat, all Jackson ImmunoResearch) in PBS for 1 h at room temperature on a shaking platform. After 5–6 washes with TBS-T over a time period of 1 h, membranes were incubated with Western Lightning Plus-ECL (Perkin Elmer) or Clarity / Clarity Max ECL Substrate (Bio-Rad) for 1 min, wrapped in foil and developed using X-ray films or with a Fusion FX chemiluminescence detector in conjunction with Evolution-Capt V18 software (Vilber).

**Production and purification of MBP-RNF219-C.** An MBP-tagged C-terminal fragment of RNF219 (residues 434-726) was produced in *Spodoptera frugiperda* Sf21 insect cells using the MultiBac baculovirus expression system as previously described[25]. In brief, the Sf21cells were grown to a density of $2 \times 10^6$ cells/ml at 27 °C in Sf900II medium (Thermo Fisher Scientific), infected with the V1 RNF219-C stock of baculovirus, and harvested 48 h after they stopped dividing. Cells were resuspended in lysis buffer (50 mM HEPES, 500 mM NaCl, pH 7.5) and lysed using a Branson Ultrasonics Sonifier SFX550. The lysate was cleared by centrifugation at 40,000 g for 1 h at 4 °C and filtered through 0.45 μm syringe-driven filters (Millipore). The cleared and filtered lysate was diluted to 250 mM NaCl before it was loaded onto a 5 ml MBPTrap column (Cytiva). The bound protein was eluted in one step with binding buffer (50 mM HEPES, 200 mM NaCl, pH 7.5) supplemented with 30 mM maltose.

**Production and purification of MBP-RNF219 434-600, 434-600 ΔSLiM and 434-600 SLiM-3E.** An MBP-tagged minimal C-terminal fragment of RNF219 (residues 434-600) was produced in *E. coli* BL21 (DE3) Star cells (Thermo Fisher Scientific) in LB medium at 20 °C as fusion protein carrying an N-terminal His₆-MBP tag and a C-terminal StrepII tag. Cells were resuspended in lysis buffer (50 mM HEPES, 500 mM NaCl, 30 mM Imidazole, pH 7.5) and lysed using a Branson Ultrasonics Sonifier SFX550. The lysate was cleared by centrifugation at 40,000 g for 1 h at 4 °C. The cleared lysate was loaded onto a 5 ml HisTrap column (Cytiva). The bound protein was eluted over a linear gradient with elution buffer (50 mM HEPES, 200 mM NaCl, 500 mM Imidazole, pH 7.5). The protein was then diluted to 75 mM NaCl and loaded onto a 5 ml HiTrap Q (Cytiva) column to be purified with anion exchange chromatography. In addition, the two mutated constructs, RNF219 ΔSLiM (residues 434-600 with deletion of 540-549) and RNF219 SLiM-3E (residues 434-600 with three mutations: L541E, M544E, M545E) were produced and purified as the wild-type version.

**StrepTactin pull-down assay.** StrepII-tagged MBP, MBP-tagged RNF219-C (residues 434-726), MBP-tagged RNF219 434-600, MBP-tagged RNF219 434-600 ΔSLiM and MBP-tagged RNF219 434-600 SLiM-3E were produced in *E. coli* BL21 (DE3) Star cells (Thermo Fisher Scientific) grown in auto-induction medium overnight at 37 °C. Cells were lysed using a Branson Ultrasonics Sonifier SFX550 in a buffer containing 50 mM Tris-HCl, 100 mM KCl, pH 8.5. The cleared lysates were incubated with 50 μl of StrepTactin Sepharose High Performance resin (Cytiva). After 1 h incubation, beads were washed four times with PBS containing 0.03% (v/v) Tween-20 and resuspended in binding buffer (50 mM Tris-HCl, 100 mM KCl, pH 8.5). Purified modules of the human CCR4-NOT complex (40 μg) as previously described[25,90] were added to the bead-bound proteins. After 1 h incubation, beads were washed three times with binding buffer and proteins were eluted with 5 mM biotin in binding buffer. The eluted proteins were analyzed by SDS polyacrylamide gel electrophoresis followed by Coomassie blue staining.

**In vitro ubiquitination assay.** Purification of human UBA1 (E1), UbcH5b, UbcH5c, UbcH10, Cdc34 (E2s) and His-ubiquitin were previously described[91–93]. Immuno-purified 3xFS-RNF219 or CCR4-NOT complex was incubated with 75 μM ubiquitin, 0.17 μM E1 enzyme and 1 μM E2 enzyme in in vitro ubiquitination buffer (30 mM Tris-HCl pH 7.5, 100 mM NaCl, 1 mM DTT, 5 mM MgCl₂) supplemented with 5 mM ATP or an equal amount of solvent for 2 h at 32 °C. The reaction was stopped by addition of 2x SDS sample buffer.

**In vitro deadenylation assay.** Deadenylation assays were performed under similar conditions as previously described[25,45,68]. In brief, MBP alone, MBP-tagged *Dm* Bam CBM, MBP-tagged RNF219-C, MBP-tagged RNF219 434-600, MBP-tagged RNF219 434-600 ΔSLiM and MBP-tagged RNF219 434-600 SLiM-3E (all at 2250 nM) were incubated with CCR4-NOT_MINI (50 nM) or NOT6:NOT7 (250 nM) for 15 min at 4 °C before mixing with a 5′-fluorescein-labeled UCUAAAU(A)₂₀ substrate RNA (50 nM). The deadenylation assay was carried out at 37 °C in a buffer containing 20 mM PIPES pH 6.8, 10 mM KCl, 40 mM NaCl, and 2 mM Mg(OAc)₂ and stopped at regular time intervals by mixing with 3x reaction volume of RNA loading dye (95% [v/v] deionized formamide, 17.5 mM EDTA pH 8, 0.01% [w/v] bromophenol blue). The reaction products were resolved on a denaturing TBE-urea polyacrylamide gel and imaged using an Amersham Typhoon Biomolecular Imager (Cytiva). Densitometric quantitation of in vitro deadenylation experiments was performed using ImageQuant TL software v8.2 (Cytiva). Deadenylation rates were estimated by the slope of the linear regression of the most abundant tail length at each time point[68].

**Electrophoretic mobility shift assay (EMSA).** Binding reactions contained 50 nM of labeled RNA and increasing concentrations of RNF219-C in a total reaction volume of 15 μl of binding buffer (20 mM PIPES pH 6.8, 40 mM NaCl, 10 mM KCl, 2 mM Mg(OAc)₂, 3% [w/v] Ficoll 400, 0.05% [v/v] NP-40, 0.03% [w/v] Orange G). The proteins were incubated for 20 min with RNA at 37 °C. The RNA-protein complexes were analyzed by electrophoresis on two gel types: 6% nondenaturing polyacrylamide gel at 10 V/cm and 2% TBE/agarose gel at 10 V/cm. Gels were imaged using an Amersham Typhoon Biomolecular Imager (Cytiva).

**Structure prediction using AlphaFold2**. We performed NOT9:RNF219 complex structure prediction using AlphaFold[69] as implemented in ColabFold[70]. A sequence where NOT9, residues 19-285, was concatenated at the C-terminal end to RNF219, residues 424-726, via a 30-glycine linker was used as input in default settings to generate five models. No additional templates or constraints were employed.

**mRNA decay assay**. HeLa cells were washed once with pre-warmed PBS and trypsinized. Cells were then resuspended in regular growth medium containing 5 μg/ml actinomycin D (AppliChem), and harvested at three (0, 3, 6 h) or four consecutive time points (0, 2, 4, 6 h). Culture medium was aspirated, and cells were immediately flash frozen in liquid nitrogen. RNA isolation was performed using the Universal RNA Purification Kit (EURx/Roboklon) including an on-column DNase digestion step. The RNA was analyzed by northern blotting, qRT-PCR or RNA-Seq as described below. mRNA half-lives were calculated assuming first-order decay kinetics. mRNA abundance was normalized to 18S rRNA and plotted against time. Curves with the following equation were fitted to the data points by linear regression: $y = a \times e^{(b \times t)}$, where y stands for the relative mRNA signal and t for the time. mRNA half-lives were calculated as follows: $t_{1/2} = \ln(2) / -b$.

**RNA-Seq**. Prior to rRNA depletion using the Ribo-Zero Plus rRNA Depletion Kit (Illumina), 1 μg total RNA was mixed with 2 μl of 1:100 diluted ERCC RNA Spike-In Mix (Thermo Fisher Scientific). Libraries were prepared with the NEBNext Ultra II Directional RNA Library Kit (NEB). Samples were equimolarly pooled and sequenced on a NextSeq550 system equipped with NextSeq System Suite v2.2.0 (Illumina) with 75 bases single end.

Reads were mapped with STAR v2.5.3a[94] to a common reference of the human genome (hg38) and the ERCC RNA Spike-In Mix, providing the basic set of Gencode V27 as downloaded from the UCSC Genome Browser wgEncodeGencodeBasic27 table as transcript annotations, allowing up to 2 mismatches and chimeric read detection with a minimum of 10 nt per segment (–chimSegmentMin 10). Read counts were summarized at the gene level with the featureCounts function of the subread package v1.6.3[95]. A read was only counted when it was contained entirely within an exon. In order to calculate half-lives with an in-house-developed R script, read counts were divided by the sum of all reads assigned to the spike-in sequences. Linear regression was performed on log-transformed read counts. Half-lives were calculated from the slope of the regression line with the equation: $\ln(2) / -$slope. Only genes with a positive half-life and a coefficient of determination $R^2 > 0.5$ in all conditions were used for subsequent analyses. For categorization, mRNAs that are transcribed from immediate early genes[75], or are bound by YTHDF2[72], TTP/ZFP36[73] or Roquin/RC3H1[74] were grouped.

**RNA gel electrophoresis and northern blotting**. 3–10 μg total RNA was mixed with 2x MOPS loading buffer (40 mM MOPS pH 7.0, 1 mM NaAc, 10 mM EDTA, 51.4% [v/v] formamide, 6.8% [v/v] formaldehyde, 7.1% [v/v] glycerol, 50 μg/ml ethidium bromide, bromophenol blue) and denatured for 10 min at 65 °C. RNA was resolved by size using 1.1% or 1.6% agarose/2% [v/v] formaldehyde/MOPS (20 mM MOPS pH 7.0, 0.5 mM NaAc, 1 mM EDTA) gels. Loading and electrophoresis were examined under UV light and RNA was blotted over night with 8x SSC buffer (1.2 M NaCl, 120 mM sodium citrate) onto Hybond-N+ nylon membranes (GE Healthcare). RNA was immobilized on membranes by UV crosslinking at 254 nm with 120 mJ/cm² twice. Membranes were briefly rinsed in 2x SSC buffer and transferred to hybridization glass tubes. Detection of poly(A) RNA, *globin* and *nucleolin* mRNAs was performed as previously described[2,53]. For the detection of poly(A) RNA, pre-hybridization was performed with 5 ml of hybridization buffer (50% [v/v] formamide, 5x SSC, 0.1% [w/v] Ficoll 400, 0.1% [w/v] poly-vinylpyrrolidone, 0.1% [w/v] BSA fraction V, 5 mM EDTA, 10 mM PIPES pH7.0, 0.4 mg/ml yeast RNA, 1% [w/v] SDS) for 30 min at 30 °C. For the detection of *globin* and *nucleolin* mRNAs, pre-hybridization was performed with 10 ml of hybridization buffer for 30 min at 55 °C. DIG-labeled oligo d(T)₁₈, globin and nucleolin probes (500 pmol) were denatured for 5 min at 95 °C and added to pre-hybridized membranes. Hybridization was performed in a rotation oven over night at 30 or 55 °C, respectively. For the detection of poly(A) RNA, membranes were washed twice with pre-warmed 2x SSC/0.1% [w/v] SDS for 5 min at 30 °C and twice with 0.1x SSC/0.1% [w/v] SDS for 15 min at room temperature. For the detection of *globin* and *nucleolin* mRNAs, membranes were washed twice with pre-warmed 2x SSC/0.1% [w/v] SDS for 5 min at 65 °C and twice with 0.5x SSC/0.1% [w/v] SDS for 20 min at 65 °C. Membranes were briefly rinsed in DIG wash buffer (10 mM maleic acid, 15 mM NaCl, 0.03% [v/v] Tween-20, pH 7.5) and blocked with Northern Blot Blocking Solution (Roche) supplemented with Maleic Acid buffer (10 mM maleic acid, 15 mM NaCl, pH 7.5) for 30 min at room temperature on a shaking platform. Alkaline phosphatase-coupled α-DIG antibodies (Roche) were added to the blocking solution at 1:5,000 and membranes were further incubated for 30 min at room temperature prior to washing four times 15 min with DIG wash buffer. For detection, membranes were briefly rinsed in DIG detection buffer (130 mM Tris-HCl pH 9.5, 100 mM NaCl) and incubated for 5 min with CDP-Star Nucleic Acid Chemiluminescence Reagent (Perkin Elmer) diluted 1:10 in DIG detection buffer. Membranes were wrapped in foil and developed using X-ray films or a Fusion FX chemiluminescence detector in conjunction with Evolution-Capt V18 software (Vilber).

**RT-qPCR**. After on-column DNase digestion, 1 μg total RNA was reverse transcribed with 100 U M-MLV Reverse Transcriptase (Promega) in the presence of 4 μM Random Hexamer Primer (Thermo Fisher Scientific), 20 U RNasin Ribonuclease Inhibitor (Promega) and dNTPs (1 mM each) for 1 h at 37 °C. PCR reactions were assembled in 384-well plates using a 5-fold diluted cDNA reaction, 400 nM of each gene-specific primer and the PowerUp SYBR Green Master Mix (Thermo Fisher Scientific) in a final volume of 10 μl per well. Quantitative PCR was performed on a QuantStudio 5 system (Thermo Fisher Scientific). Triplicates were measured for every target/reference gene. To verify removal of genomic DNA and to avoid genomic signals, all measurements included negative controls where the reverse transcriptase was omitted from the cDNA reaction. To measure the efficiency of individual primer pairs, dilution series of one cDNA sample were prepared to generate a standard curve. Gene-specific primer sequences used for the detection of mRNAs were designed with the Universal Probe Library Assay Design Center (Roche), synthesized by Eurofins Genomics, and are listed in Supplementary Data 4.

**Plasmid construction**. The following plasmids have been previously described: pcDNA3-HA (p2003)[96], pTOPuro (p2433) and pTOPuro-mycStrep (p2484)[47], puroMXβ-CDE(O)₃₇-V3 (p2823)[2], pLNCX2-EGFP-Roquin (p2838)[2,97], pcDNA3-HA-PP7cp-CAF1a-WT (p2742) and pcDNA3-HA-PP7cp-CAF1a-K4A (p3241)[53], pnYC-vHM[37,98] and pLIB[99]. PWPI-Blr (p3784), pCMV-Δ8.91 (p3785) and pMD2.G (p3786) were kindly provided by Alessia Ruggieri (Center for Integrative Infectious Disease Research, University Hospital Heidelberg). pMT2SM-HA-CNOT9 (p2704) was kindly provided by Marc Timmers (Medical Center, University of Freiburg). pSpCas9(BB)-2A-GFP (p3511) was a gift from Feng Zhang (Broad Institute, Massachusetts Institute of Technology, Addgene plasmid #48138). AAVS1-Puro-PGK1-3xFLAG-2xStrep (p3513) was kindly provided by Yannick Doyon (Center Hospitalier Universitaire de Québec Research Center-Université Laval, Addgene plasmid #68375).

For TOPuro-CNOT1-gDNA (p3515), part of human *CNOT1* flanking exon 2 was PCR amplified with primers G3983/G3984 from gDNA and inserted into pcDNA4/TO (p2430, Thermo Fisher Scientific) via BamHI/XhoI.

The *CNOT1* repair template was assembled as previously described[100]. To generate the 3xFLAG-2xStrep-*CNOT1*-repair template (p3528), the left homology arm was PCR amplified with primers G4090/G4091 from gDNA and inserted into plasmid p3513 via NdeI/NcoI. The right homology arm was PCR amplified with primers G4259/G4260 from plasmid p3515 and inserted via BstBI/EcoRI. To generate the 3xFLAG-2xStrep-*CNOT1*-repair template containing mutated PAMs (p3598), oligos G4502/G4503 were annealed and inserted into p3528 via EagI/BamHI.

Scarless cloning of guide RNA sequence oligos was performed as previously described[101]. For pSpCas9(BB)-2A-GFP-*CNOT1*-guideB (p3544) and pSpCas9(BB)-2A-GFP-*CNOT1*-guideC (p3568), oligos G4086/G4087 (guide B) or G4384/G4385 (guide C) were annealed and inserted into plasmid p3511 via BbsI sites. To generate pSpCas9(BB)-2A-GFP-*RNF219*-guideA (p3719) and pSpCas9(BB)-2A-GFP-*RNF219*-guideC (p3721), oligos G5014/G5015 (guide A) or G5018/G5019 (guide C) were annealed and inserted into plasmid p3511 via BbsI sites.

To generate pcDNA4/TO-3xFLAG-SBP-RNF219 (p3701), the 3xFLAG coding sequence was PCR amplified with primers G3987/G3988 from plasmid p3513 and ligated via NheI to the SBP coding sequence, which was amplified by PCR with primers G3990/G3991 from plasmid p2484. The ligation product was subsequently cloned into plasmid p2430 via HindIII/SacI, and the RNF219 coding sequence, amplified by PCR with primers G4972/G4973 from cDNA, was cloned into BamHI/XhoI of the resulting vector. pcDNA4/TO-3xFLAG-SBP-RNF219-RINGmut (p3709) was generated by site-directed mutagenesis of plasmid p3701 with primers G4984/G4985. Likewise, pcDNA4/TO-3xFLAG-SBP-RNF219-ΔSLiM (p3796) was generated by site-directed mutagenesis of plasmid p3701 with primers G5335/G5336.

For pWPI-3xFLAG-SBP-Stop (p3760), the 3xFLAG-SBP coding sequence was amplified by PCR with primers G5172/G5174 from plasmid p3701 and inserted into plasmid p3784 via BamHI/SmaI. For pWPI-3xFLAG-SBP-RNF219-WT (p3761), pWPI-3xFLAG-SBP-RNF219-RINGmut (p3762) and pWPI-3xFLAG-SBP-RNF219-ΔSLiM (p3802), the RNF219 coding sequence was amplified by PCR with primers G5172/G5173 from plasmid p3701 (RNF219-WT), plasmid p3709 (RNF219-RINGmut) and plasmid p3796 (RNF219-ΔSLiM), respectively, and inserted into plasmid p3784 via BamHI/SmaI.

To generate pcDNA3-HA-NOT9 (p3798), the NOT9 coding sequence was amplified by PCR with primers G5339/G5340 from plasmid p2704 and inserted into plasmid p2003 via BamHI/XhoI.

To produce TOPuro-FLAG-TEV-SBP (p3373), a NheI restriction site was introduced into p2433 via site-directed mutagenesis with primers G3483/G3484, into which annealed primers G3485/G3486 introducing a FLAG-TEV site and a PCR product coding for SBP, generated with primers G3487/G3488 from p2484 as template, were consecutively inserted via XhoI/NheI and NheI/BclI, respectively. To generate TOPuro-CAF1a-WT-FLAG-TEV-SBP (p3374) and TOPuro-CAF1a-

K$_4$A-FLAG-TEV-SBP (p3376), the coding sequence of CAF1a was excised from p2742 and p3241 and inserted into p3373 via BamHI/XhoI.

To produce pnYC-vMH-RNF219-C, a synthetic gene fragment of human RNF219 encoding residues 434-726 was codon optimized for expression in *E. coli* (GeneWiz) and inserted into the pnYC-vHM plasmid via NdeI/BamHI.

To generate pLIB-vHM-RNF219-C, human RNF219 encoding residues 434-726 was PCR amplified with primers EV0625/EV0626 from a synthetic gene fragment of RNF219 codon optimized for expression in *S. frugiperda* (GenScript) and, together with a 6xHis-MBP tag, PCR amplified from plasmid pnYC-vHM with primers EV0623/EV0624, simultaneously inserted into pLIB via Gibson assembly.

To produce pnYC-pHM-RNF219 434-600, human RNF219 encoding residues 434-600 was PCR amplified with primers JC01/JC02 using pnYC-vHM-RNF219-C as template and inserted into pnYC-pHM (same as pnYC-vHM but with a 3 C protease cleavage sequence instead of the TEV sequence) via Gibson assembly.

pnYC-pHM-RNF219 434-600 ΔSLIM and pnYC-pHM-RNF219 434-600 SLIM-3E were generated by site-directed mutagenesis of pnYC-pHM-RNF219 434-600 using overlapping primers JC018/JC019 and JC020/JC021, respectively.

Mutations, cloning boundaries and coding sequences of all plasmids were verified by Sanger sequencing (Eurofins Genomics). All DNA oligonucleotides used for cloning are listed in Supplementary Data 4.

**Statistics and reproducibility**. Data are presented as mean ± SD, unless otherwise stated. The number of independent experiments, the type of statistical test and specific *p* values are indicated in the figure legends. $p < 0.05$ were considered statistically significant. All experiments were repeated independently two to six times with similar results.

Blots and gels were processed and analyzed with ImageJ v1.52[102]. Sequence alignments were generated with Clustal Omega. ChIP-Seq data were visualized with Integrative Genomics Viewer v2.9.4. High throughput data were analyzed with R v3.6.3. Statistical analysis was performed using Microsoft Excel 2019 or GraphPad Prism v8.4.1. Statistical significance was calculated by performing a two-sided, paired Student's *t* test whenever an equal number of repeats was performed for every condition. When values were calculated relative to a control treatment, control samples were set to 1 and a two-sided, one sample *t* test was performed.

**Reporting summary**. Further information on research design is available in the Nature Research Reporting Summary linked to this article.

## Data availability

The data supporting the findings of this study are available from the corresponding authors upon reasonable request. Raw and analyzed data of RNA-Seq experiments have been deposited at the GEO under the accession number GSE172019. Raw and processed mass spectrometry data have been deposited at the ProteomeXchange Consortium via the PRIDE partner repository with the dataset identifier PXD027237. ChIP-Seq data from CD4$^+$ T-cells[61] were obtained from GEO under the accession number GSE15735. Source data of experiments shown in Figs. 1b–e, 2b–e, 3b–e, 4b–f, 6b–d, 7a–c, 7e–h and Supplementary Figs. 2a–e, 3a–c, 4a–b, d–f, 5, 6b, 10a, b, 11b, c, 13a, b, 14a–g, 15a–d are provided with this paper. Source data are provided with this paper.

## Code availability

The pipeline used to calculate mRNA half-lives from RNA-Seq data has been deposited at the OSF (https://osf.io/vskje/?view_only=13961ac6d5cd4d3ba3521615cc38fe47).

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

## Acknowledgements

We would like to thank Thomas Ruppert and Sabine Merker from the ZMBH Mass Spectrometry and Proteomics Core Facility, the NGS Core Facility at the Institute of Clinical Chemistry of the Medical Faculty Mannheim, Frauke Melchior (ZMBH) for kindly providing recombinant proteins for in vitro ubiquitination assays, Stefanie Uhlig (FlowCore Mannheim, Medical Faculty Mannheim of Heidelberg University) for cell sorting, Alessia Ruggieri (Center for Integrative Infectious Disease Research, University Hospital Heidelberg) for generously providing plasmids for lentiviral transduction, Marc Timmers (Medical Center, University of Freiburg), Feng Zhang (Broad Institute, Massachusetts Institute of Technology) and Yannick Doyon (Center Hospitalier Universitaire de Québec Research Center-Université Laval) for kindly providing plasmids. We would like to acknowledge Michihito Wakai (Division of Biochemistry, Medical Faculty Mannheim of Heidelberg University; and Kyoto University) for cloning the pcDNA4TO-3xFLAG-SBP construct, Ann-Bin Shyu (McGovern Medical School, University of Texas, Houston) for kindly providing the CAF1a antibody and Katharina Hoerth (Division of Biochemistry, Medical Faculty Mannheim of Heidelberg University) for critically reading the manuscript. This work was funded by the Deutsche Forschungsgemeinschaft (DFG), project number 273941853-SPP 1935, 278001972-TRR 186 and 439669440-TRR 319 to G.S.; and the Intramural Research Program, Center for Cancer Research, National Cancer Institute, National Institutes of Health to J.C., Y.L., and E.V.

## Author contributions

G.S., E.V. and F.P. conceived the study, analyzed data, and wrote the manuscript. F.P. performed purification of the endogenous CCR4-NOT complex and carried out the majority of experiments, J.C. produced recombinant RNF219-C, carried out in vitro pull-down as well as in vitro deadenylation experiments and analyzed in vitro data, Y.L. assisted with in vitro deadenylation assays, A.S. generated FST-NOT1 HeLa cells, D.L. prepared and sequenced RNA-Seq libraries, V.M. performed nucleofection of CRISPR/Cas9 RNPs for the generation of NOT9 KO cells, S.L. assisted with the analysis of ChIP-Seq data, J. Schott performed bioinformatics analyses and J. Schweiggert produced recombinant proteins for in vitro ubiquitination assays.

## Funding

## Competing interests

The authors declare no competing interests.
