## [Peer Review File · Nature Communications]

Reviewers' Comments:

Reviewer #1:

Remarks to the Author:

The deadenylation activity of the CNOT complex is central in the precise modulation of mRNA stability and translational status. In this manuscript, the authors demonstrate the acetylation-dependent inhibition of deadenylation activity of CNOT complex by RNF219. The authors utilize IP-MS approaches to identify RNF219 as an HDAC-dependent interactor of the CNOT complex. Further, the residue in RNF219 that bind CNOT7 is characterized and the deadenylation inhibition activity is characterized by in vitro and in cell experiments.

However, it appears that the manuscript is largely a repetition of an existing publication (<https://doi.org/10.1093/jmcb/mjaa061>) at the time of submission and thus lacks novelty. Also, the mass spectrometry data seems weak to firmly support the linkage between HDAC inhibition and the association of RNF219 with the CNOT complex (P.S. the raw mass data remains inaccessible for the reviewer). In addition, the molecular mechanism underlying the dissociation of RNF219 and the CNOT complex upon HDAC inhibition – the reduction in the RNF219 expression level – seems to be misleading and vague. Collectively, I do not recommend this paper to be published in Nature Communications.

Major concerns:

1. The manuscript lacks novel findings. Data in Figs. 1-4 are a near-complete overlap with those in an aforementioned pre-existing publication (<https://doi.org/10.1093/jmcb/mjaa061>). Given the suppressive roles of RNF219 in mRNA deadenylation, the mRNA half-life analysis in Fig. 5 seems rather accessory.
2. Fig.1b, d: The mass spectrometry source data was not available for review process (neither spectral search results in supplementary table format, nor raw data depository). This way, one could not evaluate the validity of identification, quantification, nor reproducibility in mass spectrometry data. This should be made accessible for appropriate review.
3. Fig. 1d: The authors illustrate the HDAC-dependency of RNF219-CNOT association only using RMD treatment. To exclude any potential off-target effects, a rescue experiment i.e. HDAC compensation should be performed in parallel. Orthogonal approaches such as knockdown experiments, or IP with opposite proteins should also help. Of note, the term 'HDACi' for RMD treatment can be misleading for the audience, and needs to be replaced.
4. In the Results section, "To better understand the reason for reduced association results in its depletion from the CCR4-NOT complex.": The authors propose that inhibition of HDAC leads to decreased RNF219 mRNA levels, which results in less RNF219-CNOT association. However, RNF219 itself is a potent RNA stabilizer (by inhibition of CNOT complex). Post-translational depletion of RNF219 can result in the destabilization of its own mRNA. Hence, the causal directionality between the reduction of RNF219-CNOT association and the decrease of RNF219 mRNA level cannot be determined with the presented data.
5. Related to points 3-4: The authors did not provide any molecular insight into how HDAC inhibition can stabilize RNF219-CNOT association. One explanation could be transcriptional modulation of RNF219 expression via HDAC inhibition. A pre-existing study might provide some hints into the regulatory mechanism (<https://doi.org/10.1016/j.molcel.2016.08.030>).

Reviewer #2:

Remarks to the Author:

In this manuscript, Poetz et al describe the RING finger protein RNF219 as an acetylation-regulated inhibitor of the conserved CCR4-NOT complex. By tagging the NOT1 subunit using genome editing, they identify six proteins that associate with CCR4-NOT, validate two of them by western blot, and further characterize one of them (RNF219) by both cellular and in vitro assays. The finding that RNF219 associates with CCR4-NOT is not novel as several other publications have now shown this.

In a more novel result, the authors show that RNF219 protein and mRNA abundance, and its association with CCR4-NOT is reduced after treatment with HDAC inhibitors. They also show that RNF219 associates with CCR4-NOT by binding to the NOT9 subunit via a short linear motif (SLiM, or a Caf40-binding motif). By employing in vitro deadenylation assays and RNAseq, they show that RNF219 is an inhibitor of deadenylation. Furthermore, they show that RNF219 is an active E3 ubiquitin ligase by in vitro ubiquitination assays, but this E3 ligase activity is not required for its inhibition of deadenylation. Finally, they propose a model where RNF219 could inhibit deadenylation by masking an RNA-binding surface on NOT9 to displace mRNA non-specifically. As another possibility, they argue that RNF219 might affect CAF1 directly as it is proximal to the NOT9 subunit. A limitation of the study is that the biological function of RNF219 remains unclear, eg. under what circumstances does it regulate deadenylation? How is it regulated by acetylation?

Overall, this is a very well-written manuscript, which is mostly supported by the carefully conducted experiments. It describes new contributions to the field of gene expression/RNA biology, and I enjoyed reading it. That said, there are a few suggestions that should be addressed, which would make the paper stronger.

Suggestions – Major Comments:

- 1) The punchline of the paper is that RNF219 is an acetylation-driven deadenylation inhibitor. Is RNF219 itself acetylated? If not, how is it regulated by acetylation? Is regulation by acetylation dependent on CAF1 acetylation? Experimentally addressing some of the issues listed below will be beyond the scope of the present work, but in the very least, the authors should offer some speculation/discussion on how acetylation regulates RNF219 including:
 - a. Is RNF219 acetylated?
 - b. Do you still see the inhibition if CAF1 cannot be acetylated?
 - c. Is acetylation regulating something at the transcription level that changes RNF219 association or are there any other acetylation targets?
 - d. Is the inhibition direct or indirect, i.e. other accessory binders needed?
- 2) RNF219 inhibits deadenylation using in vitro assays. However, a 45-fold excess of RNF219 was used.
 - a. If RNF219 is stably associated, why use such excess of protein? Shouldn't 1:1 be sufficient?
 - b. How can the authors be certain that this excess of protein does not generate bias on its inhibition activity? For example, could RNF219 non-specifically bind and sequester substrate RNA when it is present at such high concentrations? An EMSA at similar concentrations would help clarify.
 - c. Zinc was included in the RNF219 purification. Even though it was not in the elution buffer, using 45x fold excess might introduce sufficient zinc into the deadenylation assays to affect CCR4-NOT activity (see PMID: 19307292). It would therefore be helpful to include a proper negative control with some zinc included.
 - d. On page 11, the authors describe RNF219 as a potent inhibitor of CNOT. Given the high excess of RNF219 in the deadenylation assays "potent" seems to be inappropriate to describe the observed effects. Please re-phrase.
- 3) EMSA assays would be very fast and effective to assess whether the RNF219 association with NOT9 affects interaction with RNA.
- 4) Figure 5c & e: Some of the differences between the indicated samples are not statistically significant and therefore these data do not support statements made in the text. For example, the p values are the same for parent vs KO and parent vs WT rescue. Can the authors clarify?

Minor comments:

- 1) On page 5, the authors list six proteins that they pull-down by endogenous FST-NOT1 purification. They mention that they did not further pursue CAPZA1 and CAPZAB. As a reader, it would be great if the authors can provide more information on why these two have not been further pursued, particularly because other factors may affect RNF219 activity on deadenylation.
- 2) The authors mention on page 8 that RNF219-deltaSLiM shows only a small shift to the lighter fractions compared to WT because of its association to the origin recognition complex. Could the

authors verify this by blotting for one of the proteins of the origin recognition complex?

3) Figure 2c: Listing the corresponding CNOT1 fragment in the sub-complexes used for pull-down would be beneficial for the understanding of the figure. Generally, the figure could be improved by labelling each protein in the pull-down.

4) Could the authors clarify how many repetitions were done for the quantification of the glycerol gradient in Supp. Fig. 3b

5) RNF219 E3 ligase activity seems to be strikingly stimulated after incorporation into CCR4-NOT (Fig 3c-d). Is it possible to quantify the fold activation?

6) Could the authors also add a quantification of their glycerol gradients in Supp. Fig 8 & 10?

7) Please state the concentration of RNF219 and CNOT used in the ubiquitination assays.

8) Top of page 9 – I presume this statement refers to the RNF219 SLiM and not all SLiMs that interact with NOT9? Is there any sequence conservation of RNF219 SLiM with other Caf40-binding motifs?

9) Figure S2c – what are the two bands in the northern blot?

Reviewer #3:

Remarks to the Author:

The evolutionarily conserved CCR4-NOT complex plays a vital role in controlling mRNA turnover and other aspects of mRNA metabolism both at the transcriptome level and in a transcript-specific manner. However, still little is known about how the complex is regulated and by what post-translational modifications and auxiliary factors. The Stoecklin lab showed a few years ago that protein acetylation has a profound impact on global mRNA turnover. The group showed that upon histone deacetylase (HDAC) inhibition global mRNA degradation is accelerated through CCR4-NOT complex. In this study, the authors set out addressing several related and critical issues by asking whether there may have acetylation-regulated changes in the protein interactome of the CCR4-NOT complex as well as physiological changes in the composition of the CCR4-NOT complex. Through well controlled mass-spec based purification and quantification experiments, they were able to pinpoint RNF219 as a potential auxiliary factor whose level diminishes upon RMD-elicited hyperacetylation. This in turn resulted in RNF219 diminished presence in the CCR4-NOT complex. Through a series of molecular and biochemical analysis, they identified the RNF219 interacting partner in the CCR4-NOT core complex, i.e., NOT9, and further mapped the subregions in RNF219 and NOT9 necessary for the interaction. After confirming the E3 ubiquitin ligase autoubiquitination activity, they showed that RNF219-mediated ubiquitination activity does not influence the composition of the CCR4-NOT complex or promote proteolytic degradation of core subunits. They then investigated whether RNF219 may actually modulate the deadenylation function of CCR4-NOT complex by conducting both in vitro and in vivo deadenylation assays. To determine transcriptome-wide impact of RNF219 on global mRNA decay, they generated RNF219 KO cell lines and brilliantly the isogenic lines that express either 3XFS-RNF wt or mutant for rescue experiments. These experiments showed that RNF219 protein per se and not its E3 ligase activity is required to inhibit the deadenylation function of CCR4-NOT complex and that RMD treatment induces degradation of RNF219 to lessen the inhibition.

Overall, this is a very interesting study, which reveals a novel mechanism by which CCR4-NOT complex is regulated through RNF219 and how HDAC inhibition elicits RNF219 degradation, thus activating the CCR4-NOT complex for deadenylation. The manuscript is well written with major conclusions properly drawn. The experiments and designs are well controlled and thought. The data are of strong quality. In this reviewer's opinion, all the data go into making an important story that is of broad and significant interest to general readers served by Nature Communications.

This reviewer has the following points for the authors' attention:

1. Lines 259-261: The authors showed that RNF219-C elicits a three- to four-fold reduction in the apparent deadenylation rate of the CCR4-NOTMINI complex in vitro. They then concluded that "RNF219 acts as a potent inhibitor of CCR4-NOT-mediated deadenylation in vitro". This conclusion should be toned down as the in vivo demonstration that complements the in vitro results has not yet shown at this point. Alternatively, the authors may consider testing the intact RNF219 instead of the C-terminal portion in the experiment described in Fig. 4b
2. Lines 303-305: The authors concluded that "RNF219 is a general inhibitor of the CCR4-NOT deadenylase complex and this effect may be more pronounced for short-lived transcripts". The second statement would be more convincing if the authors can show the comparisons of stability changes between KO and WT cells in subgroups with different half-life ranges, e.g., transcripts with half-lives <6h (short-lived), transcripts with half-lives between 6h-12h (somewhat unstable/stable), and transcripts with half-lives >12h (long-lived) and that the short-lived subgroup is indeed affected the most among the three subgroups in the comparisons.
3. Supplementary Fig. 11b-d, transcripts used in each of these boxplots of half-life calculations for YTHDF2, TTP and Roquin proteins should be given as a supplementary table and references should be cited as well.
4. Line 324, "...the BTG/TOB family, which recruit mRNAs by associating with the cytoplasmic poly(A)-binding protein PABPC11,50,72." The TOB work (ref 54) should be cited here as well. The three references cited are only about BTG proteins that lack the typical PAM2 motif for PABP interaction.
5. In the second paragraph of Discussion (lines 331-344), the authors discussed the findings concerning RNF219 by other studies. The authors may consider mentioning that RNF219 was also shown by Chen et al. (RNA 26: 1143, 2020; see Supplemental Table S3) to be among the high-confidence proteins co-purified along with CCR4-NOT complex when using TOB2, a strong CAF1-interacting protein, as the bait. One possibility is that TOB2 may somehow counteract the inhibitory effect of RNF219 when joining the CCR4-NOT-RNF219 complex, given the close proximity of the NOT9 module to the CAF1-catalytic module which TOB2 binds to strongly.

Response to reviewers:

Reviewer #1:

The deadenylation activity of the CNOT complex is central in the precise modulation of mRNA stability and translational status. In this manuscript, the authors demonstrate the acetylation-dependent inhibition of deadenylation activity of CNOT complex by RNF219. The authors utilize IP-MS approaches to identify RNF219 as an HDAC-dependent interactor of the CNOT complex. Further, the residue in RNF219 that bind CNOT7 is characterized and the deadenylation inhibition activity is characterized by in vitro and in cell experiments.

However, it appears that the manuscript is largely a repetition of an existing publication (<https://doi.org/10.1093/jmcb/mjaa061>) at the time of submission and thus lacks novelty. Also, the mass spectrometry data seems weak to firmly support the linkage between HDAC inhibition and the association of RNF219 with the CNOT complex (P.S. the raw mass data remains inaccessible for the reviewer). In addition, the molecular mechanism underlying the dissociation of RNF219 and the CNOT complex upon HDAC inhibition – the reduction in the RNF219 expression level – seems to be misleading and vague. Collectively, I do not recommend this paper to be published in Nature Communications.

Major concerns:

1. The manuscript lacks novel findings. Data in Figs. 1-4 are a near-complete overlap with those in an aforementioned pre-existing publication (<https://doi.org/10.1093/jmcb/mjaa061>). Given the suppressive roles of RNF219 in mRNA deadenylation, the mRNA half-life analysis in Fig. 5 seems rather accessory.

> Response: We agree that RNF219 has already been described as a negative regulator of CCR4-NOT-mediated deadenylation by Du et al. (2020, reference 85 in our manuscript). While this paper focuses on the role of RNF219 in stem cell differentiation, our study has a different focus and goes beyond the findings of Du et al. (2020) by providing detailed structural insight into the interaction of RNF219 with the CCR4-NOT complex, by reconstituting the inhibitory function of RNF219 in vitro in presence of a recombinant six-subunit CCR4-NOT complex, by providing a transcriptome-wide analysis of RNF219-dependent changes in mRNA half-lives, and by exploring the role of RNF219 in the context of acetylation-induced mRNA turnover.

Our study provides substantial new insight into the function and mechanism of RNF219, clearly exceeding the current state of the field. Of specific novelty are the following findings:

- 1. RNF219 is an acetylation-regulated interactor of the CCR4-NOT complex (Fig. 1c, d, e), which dissociates from the complex as a consequence of transcriptional downregulation upon HDAC inhibition (new Supplementary Fig. 4).*
- 2. FHL2 as well as the two actin associated proteins CAPZA1 and CAPZB are stable interactors of the CCR4-NOT complex (Fig. 1b, new data in Fig. 1c). This observation will allow for further investigations on how these uncharacterized binding partners affect CCR4-NOT functions.*
- 3. The binding of RNF219 to CCR4-NOT occurs via the NOT9/CAF40 module of the complex (Fig. 2b, c). This interaction strictly depends on the presence of a short linear motif (SLiM) embedded within the C-terminal low-complexity region of RNF219 (Fig. 2d, new Fig. 6b).*

4. *The C-terminal half of RNF219 is sufficient to attenuate deadenylation in vitro. This negative regulation strictly depends on the presence of NOT9 (new Fig. 4d, e) and the SLiM (new Fig. 6c, d).*
5. *RNF219 is a CCR4-NOT-associated E3 ubiquitin ligase, which is active within native CCR4-NOT complexes. However, the RNF219 E3 ligase activity is not regulated by its association with CCR4-NOT (Fig. 3c, new Fig. 3d).*
6. *The RNF219 SLiM likely adopts the fold of an amphipathic alpha-helix, which engages the concave surface of NOT9/CAF40 (new Fig. 5b, new Supplementary Fig. 12a). Structure-guided mutagenesis of hydrophobic amino acids within the SLiM disrupts the NOT9-RNF219 interaction (new Fig. 6b) and alleviates the inhibitory function of a minimal RNF219 construct on CCR4-NOT-mediated deadenylation in vitro (new Fig. 6c, d). Hence, binding of the RNF219 SLiM to NOT9 displays structural similarity with other reported CAF40-binding motifs of Drosophila Bam and Roquin (new Fig. 5a, new Supplementary Fig. 12b), representing an example of convergent evolution.*
7. *RNF219 acts as an inhibitor of global mRNA turnover (Fig. 7b, c). While the E3 ligase activity is dispensable for the stabilization of short-lived transcripts (new Fig. 7d; Fig. 7e), the stabilization of long-lived mRNAs depends on the integrity of its N-terminal RING domain (new Fig. 7d).*
8. *Reduced association of RNF219 with CCR4-NOT upon RMD treatment contributes to acetylation-induced mRNA turnover (Fig. 7g, h). Moreover, the RNF219-mediated inhibition of mRNA decay is independent of the acetylation status of the exoribonuclease CAF1a (new Supplementary Fig. 15).*

2. Fig. 1b, d: The mass spectrometry source data was not available for review process (neither spectral search results in supplementary table format, nor raw data depository). This way, one could not evaluate the validity of identification, quantification, nor reproducibility in mass spectrometry data. This should be made accessible for appropriate review.

> *Response: Raw and processed mass spectrometry proteomics data have now been deposited at the ProteomeXchange Consortium via the PRIDE partner repository with the dataset identifier PXD027237. The data is accessible with the following reviewer account details:*

Username: reviewer_pxd027237@ebi.ac.uk
Password: dmL7Itoh

3. Fig. 1d: The authors illustrate the HDAC-dependency of RNF219-CNOT association only using RMD treatment. To exclude any potential off-target effects, a rescue experiment i.e. HDAC compensation should be performed in parallel. Orthogonal approaches such as knockdown experiments, or IP with opposite proteins should also help. Of note, the term ‘HDACi’ for RMD treatment can be misleading for the audience, and needs to be replaced.

> *Response: To exclude potential off-target effects of RMD treatment, we now performed an analysis of RNF219 mRNA, pre-mRNA and protein expression following treatment with two additional, hydroxamic acid-based pan-HDAC inhibitors, Trichostatin A (TSA) and suberoylanilide hydroxamic acid (SAHA). All three HDAC inhibitors significantly reduce both RNF219 mRNA and pre-mRNA levels (new Supplementary Figure 4a) and elicit a ~2-fold reduction of RNF219 protein expression (new Supplementary Figure 4b). Hence, reduced*

expression of RNF219 can be observed following treatment with different HDAC inhibitors, which downregulate RNF219 transcription as evident from reduced pre-mRNA levels.

To gain more insight into the HDAC-dependent transcriptional regulation of RNF219, we made use of a ChIP-Seq dataset derived from CD4⁺ T-cells (<https://doi.org/10.1016/j.cell.2009.06.049>), which revealed occupancy of HDAC1 and RNA polymerase II (PolII) across the first exon of RNF219 (new Supplementary Fig. 4c). Moreover, we performed siRNA-mediated knock-down of HDAC1 in HeLa cells and found this to reduce both RNF219 pre-mRNA and protein levels (new Supplementary Fig. 4d, e). This result confirms what we observed with pharmacological inhibition of class I HDACs and provides evidence that HDAC1 has a stimulating effect on RNF219 transcription.

The term "HDACi" has been replaced with "pharmacological HDAC inhibition" throughout the text.

4. In the Results section, “To better understand the reason for reduced association results in its depletion from the CCR4-NOT complex.”: The authors propose that inhibition of HDAC leads to decreased RNF219 mRNA levels, which results in less RNF219-CNOT association. However, RNF219 itself is a potent RNA stabilizer (by inhibition of CNOT complex). Post-translational depletion of RNF219 can result in the destabilization of its own mRNA. Hence, the causal directionality between the reduction of RNF219-CNOT association and the decrease of RNF219 mRNA level cannot be determined with the presented data.

> Response: Although the majority of poly(A)-containing RNAs is rapidly degraded upon pharmacological HDAC inhibition or HDAC1/2 knock-down (<https://doi.org/10.1016/j.molcel.2016.08.030>), RNF219 mRNA degradation was found to be mostly unaffected by RMD treatment (new Supplementary Fig. 4f). Together with our RNF219 pre-mRNA/mRNA and protein expression analysis upon exposure to different HDAC inhibitors (new Supplementary Fig. 4a, b), this result demonstrates that the reduced binding of RNF219 to the CCR4-NOT complex upon HDAC inhibition is primarily due to transcriptional regulation.

To further explore whether RNF219 autoregulates its own mRNA post-transcriptionally, we examined the degradation of RNF219 mRNA in two CNOT9 knock-out clones (Reviewer Fig. 1a, below). Since the permanent loss of NOT9 expression prevents RNF219 from binding to the CCR4-NOT complex (Fig. 2b), this system allowed us to assess the impact of RNF219 on the stability of its own mRNA. For comparison, RNF219 mRNA degradation was assessed in two CRISPR control clones and parental HeLa cells. Although RNF219 acts as a general inhibitor of deadenylation and mRNA decay (Fig. 4b, c; Fig. 7b, c), we did not observe differences in the degradation of RNF219 mRNA upon loss of NOT9 (Reviewer Fig. 1b, below). Hence, there is no experimental evidence supporting post-transcriptional autoregulation of RNF219 mRNA.

Taken together, our results demonstrate that the reduced incorporation of RNF219 into the CCR4-NOT complex upon pharmacological HDAC inhibition is directly linked to its reduced expression due to transcriptional suppression.

5. Related to points 3-4: The authors did not provide any molecular insight into how HDAC inhibition can stabilize RNF219-CNOT association. One explanation could be transcriptional modulation of RNF219 expression via HDAC inhibition. A pre-existing study might provide some hints into the regulatory mechanism (<https://doi.org/10.1016/j.molcel.2016.08.030>).

> *Response: In the present study we provide evidence that pharmacological HDAC inhibition with RMD reduces the association of RNF219 with the CCR4-NOT complex (Fig. 1c, d, e). Since pharmacological HDAC inhibition and HDAC1 knock-down result in decreased RNF219 pre-mRNA levels (new Supplementary Fig. 4), the acetylation-regulated association of RNF219 with the CCR4-NOT complex upon HDAC inhibition is primarily due to transcriptional suppression of RNF219.*

Reviewer #2 (Remarks to the Author):

In this manuscript, Poetz et al describe the RING finger protein RNF219 as an acetylation-regulated inhibitor of the conserved CCR4-NOT complex. By tagging the NOT1 subunit using genome editing, they identify six proteins that associate with CCR4-NOT, validate two of them by western blot, and further characterize one of them (RNF219) by both cellular and in vitro assays. The finding that RNF219 associates with CCR4-NOT is not novel as several other publications have now shown this.

In a more novel result, the authors show that RNF219 protein and mRNA abundance, and its association with CCR4-NOT is reduced after treatment with HDAC inhibitors. They also show that RNF219 associates with CCR4-NOT by binding to the NOT9 subunit via a short linear motif (SLiM, or a Caf40-binding motif). By employing in vitro deadenylation assays and RNAseq, they show that RNF219 is an inhibitor of deadenylation. Furthermore, they show that RNF219 is an active E3 ubiquitin ligase by in vitro ubiquitination assays, but this E3 ligase activity is not required for its inhibition of deadenylation. Finally, they propose a model where RNF219 could inhibit deadenylation by masking an RNA-binding surface on NOT9 to displace mRNA non-specifically. As another possibility, they argue that RNF219 might affect CAF1 directly as it is proximal to the NOT9 subunit. A limitation of the study is that the biological function of RNF219 remains unclear, eg. under what circumstances does it regulate deadenylation? How is it regulated by acetylation?

Overall, this is a very well-written manuscript, which is mostly supported by the carefully conducted experiments. It describes new contributions to the field of gene expression/RNA biology, and I enjoyed reading it. That said, there are a few suggestions that should be addressed, which would make the paper stronger.

Suggestions – Major Comments:

1) The punchline of the paper is that RNF219 is an acetylation-driven deadenylation inhibitor. Is RNF219 itself acetylated? If not, how is it regulated by acetylation? Is regulation by acetylation dependent on CAF1 acetylation? Experimentally addressing some of the issues listed below will be beyond the scope of the present work, but in the very least, the authors should offer some speculation/discussion on how acetylation regulates RNF219 including:

a. Is RNF219 acetylated?

> *Response: RNF219 acetylation has been reported to increase upon treatment with the pan-HDAC inhibitor Trichostatin A (TSA) in RAW264.7 cells (<https://doi.org/10.1111/bph.15060>) and 5 lysines have been identified to be modified by acetylation in high-throughput mass spectrometry approaches listed on phosphosite.org. To assess the acetylation status of RNF219 in HeLa cells upon RMD treatment, we immunoprecipitated the endogenous protein followed by anti-acK western blotting (Reviewer Fig. 2, below). Since RMD treatment reduces RNF219 expression (Fig. 1c, new Supplementary Fig. 4a, b), the IP was performed under saturating conditions to capture an equal quantity of protein in the DMSO- and RMD-treated sample. While the IP efficiently enriched RNF219 in comparison to an IgG control, we could not detect a specific acetylation signal neither in the DMSO- nor in the RMD-treated condition (Reviewer Fig. 2). Although it cannot be excluded that RNF219 undergoes sub-stoichiometric acetylation at specific sites only detectable by protein MS, our current data indicate that RMD treatment does not regulate RNF219 via direct acetylation, but rather by downregulating its expression through transcriptional repression.*

b. Do you still see the inhibition if CAF1 cannot be acetylated?

> *Response: To investigate whether the RNF219-mediated inhibition of mRNA deadenylation and decay depends on the acetylation of the exoribonuclease CAF1a, we made use of a CAF1a-K₄A mutant where 4 C-terminal lysine residues are replaced with alanine. We have previously shown that acetylation of this mutant is substantially reduced in comparison to CAF1a-WT, and that CAF1a-K₄A is less active than CAF1a-WT (<https://doi.org/10.1016/j.molcel.2016.08.030>; herein Fig. 4B and 4C).*

To determine whether CAF1a acetylation is important for RNF219-mediated inhibition of mRNA decay, we now measured the stability of a constitutive decay element (CDE)-containing β -globin reporter mRNA (<https://doi.org/10.1016/j.cell.2013.04.016>) following transient co-expression of Roquin and the RNF219-RING mutant together with either CAF1a-WT or CAF1a-K4A in HeLa cells. Northern blotting showed that deadenylation and decay of the β -globin reporter mRNA is attenuated upon co-expression of the RNF219-RING mutant in presence of CAF1a-WT (new Supplementary Fig. 15b, c, d). As expected, the CAF1a-K4A mutant itself had an inhibitory effect on reporter mRNA deadenylation and decay, and co-expression of the RNF219-RING mutant had a synergistic effect by further inhibiting β -globin reporter mRNA degradation (new Supplementary Fig. 15b, c, d). Taken together, this experiment provides evidence that the RNF219-mediated attenuation of mRNA deadenylation and decay is independent of the CAF1a acetylation status.

c. Is acetylation regulating something at the transcription level that changes RNF219 association or are there any other acetylation targets?

> Response: RNF219 protein and mRNA expression were found to be reduced upon RMD treatment (Fig. 1c). To exclude any potential off-target effects of RMD treatment we performed an analysis of RNF219 mRNA, pre-mRNA, and protein expression upon exposure to two additional pan-HDAC inhibitors, Trichostatin A (TSA) and suberoylanilide hydroxamic acid (SAHA). Treatment with each of the three HDAC inhibitors led to a significant reduction of RNF219 pre-mRNA and mRNA levels, and reduced RNF219 protein expression by ~2-fold (new Supplementary Fig. 4a, b). From this we concluded that pharmacological HDAC inhibition downregulates RNF219 expression primarily due to transcriptional repression.

Analysis of HDAC1 occupancy at the RNF219 locus using a pre-existing ChIP-Seq dataset (<https://doi.org/10.1016/j.cell.2009.06.049>) revealed HDAC1 binding across the first exon of RNF219 in CD4⁺ T-cells (new Supplementary Fig. 4c). To investigate if HDAC1 is involved in regulating RNF219 expression, we performed HDAC1 knock-down experiments using siRNAs, and found that depletion of HDAC1 reduced RNF219 pre-mRNA, mRNA and protein expression (new Supplementary Fig. 4d and 4e). These results provide experimental evidence that HDAC1 enhances transcription of RNF219, and further validate our results from pharmacological HDAC inhibition.

d. Is the inhibition direct or indirect, i.e. other accessory binders needed?

> Response: Binding of RNF219 to the CCR4-NOT complex strictly depends on the presence of NOT9 (Fig. 2b), which establishes a direct interaction with the SLiM embedded within the C-terminal half of RNF219 (Fig. 2d; new Fig. 6b). In vitro, RNF219-mediated inhibition of deadenylation can only be observed with the CCR4-NOT_{MINI} complex containing the NOT9 module, but not with the nuclease module alone (new Fig. 4d, e). To gain more insight into the RNF219-NOT9 interaction, we predicted the structure of the RNF219 SLiM together with NOT9 using AlphaFold2 (new Supplementary Fig. 12a). This analysis revealed that the RNF219 SLiM likely folds into an amphipathic alpha-helix, which engages the concave surface of NOT9/CAF40 (new Fig. 5). Deletion of the SLiM as well as structure-guided mutagenesis of hydrophobic residues within the SLiM not only abrogate the RNF219-NOT9 interaction (new Fig. 6b), but also alleviate the inhibitory effect of RNF219 on deadenylation

(new Fig. 6c, d). These results provide experimental evidence that the inhibitory effect of RNF219 on deadenylation depends on the direct interaction of the NOT9 module and the C-terminal SLiM of RNF219.

2) RNF219 inhibits deadenylation using in vitro assays. However, a 45-fold excess of RNF219 was used.

a. If RNF219 is stably associated, why use such excess of protein? Shouldn't 1:1 be sufficient?

> Response: Our new data in Fig. 4f show a titration experiment with decreasing concentrations of RNF219-C. Inhibition of CCR4-NOT_{MINI}-mediated deadenylation is already apparent at the lowest concentration tested (0.05 μM), which is equimolar to the substrate and CCR4-NOT complex concentration in the assay. Hence, RNF219 efficiently inhibits the CCR4-NOT-mediated deadenylation in our in vitro assays already at much lower concentrations.

b. How can the authors be certain that this excess of protein does not generate bias on its inhibition activity? For example, could RNF219 non-specifically bind and sequester substrate RNA when it is present at such high concentrations? An EMSA at similar concentrations would help clarify.

> Response: As suggested by the reviewer, we performed EMSAs with up to 5 μM of RNF219-C using the same substrate RNA as in the deadenylation assays. Analysis by both polyacrylamide-based and agarose-based gel electrophoresis did not reveal complex formation between RNF219-C and the substrate RNA at any of the concentrations tested, whereas PABPC1 – used as a positive control for binding poly(A)-containing RNA substrates – showed robust complex formation at 0.1 μM (new Supplementary Fig. 11b and 11c). From this we concluded that RNF219 does not sequester RNA in these assays.

c. Zinc was included in the RNF219 purification. Even though it was not in the elution buffer, using 45x fold excess might introduce sufficient zinc into the deadenylation assays to affect CCR4-NOT activity (see PMID: 19307292). It would therefore be helpful to include a proper negative control with some zinc included.

> Response: We did not include zinc in the purification of recombinant RNF219. Moreover, we now demonstrate that MBP-RNF219-C does not inhibit deadenylation by the catalytic NOT6/NOT7 module in the absence of the NOT9 module (new Fig. 4d). Likewise, inhibition of CCR4-NOT-mediated deadenylation by RNF219 434-600 is completely eliminated by deletion of the SLiM or by the 3E substitution (new Fig. 6c). This confirms that purification conditions and buffers used to produce RNF219 do not impact appreciably on deadenylase activity in the in vitro assays.

d. On page 11, the authors describe RNF219 as a potent inhibitor of CNOT. Given the high excess of RNF219 in the deadenylation assays “potent” seems to be inappropriate to describe the observed effects. Please re-phrase.

> *Response: We now omitted the term "potent" in our description of RNF219 as an inhibitor of the CCR4-NOT deadenylase complex. The sentence now reads: "Hence, RNF219 has the capacity to directly inhibit CCR4-NOT-mediated deadenylation in vitro." (line 297/298)*

3) EMSA assays would be very fast and effective to assess whether the RNF219 association with NOT9 affects interaction with RNA.

> *Response: The predicted structural similarity between binding of the RNF219 SLiM, and binding of the Drosophila Roquin and Bam SLiMs to the positively charged, concave surface of NOT9 (new Fig. 5 and new Supplementary Fig. 12b) may indeed be consistent with a simple model where RNF219 competes with RNA for the same surface on NOT9. However, the situation seems more complex as RNF219 appears to mediate its inhibitory effect through two distinct mechanisms, one that is independent of the RING domain (as seen for short-lived mRNAs and in the bulk mRNA measurement; Fig. 7e and c), and one that is dependent on the RING domain (as seen for long-lived mRNAs; new Fig. 7d). We feel that unraveling the complexity of this regulation cannot be solved easily (e.g. by EMSAs alone) as it may also involve the competition of RNF219 with RBPs that interact with NOT9. We hope the reviewer understands that time did not permit to pursue this question in the context of this revision.*

4) Figure 5c & e: Some of the differences between the indicated samples are not statistically significant and therefore these data do not support statements made in the text. For example, the p values are the same for parent vs KO and parent vs WT rescue. Can the authors clarify?

> *Response: This comment now pertains to Fig. 7c and 7e. Due to the high number of $n = 1909$, the calculated p-values are all very low (hence "significant"). For simplicity, the p-values for parental vs. KO and parental vs. WT rescue are indicated as <0.001 but are of course not identical (e.g. $p = 1.2 \times 10^{-240}$ for parental vs. KO; $p = 4.1 \times 10^{-28}$ for parental vs. WT rescue in Fig. 7c). The only p-value which is below the threshold for statistical significance was calculated for parental vs. RINGmut in Fig. 7e ($p = 0.06$).*

What is more important here is to look at the effect size, e.g. by comparing the median values. For bulk mRNA half-lives in Fig. 7c, the median values are virtually the same for parental, WT and RINGmut, while the median in the KO is clearly lower. The same is true for the IEG mRNAs (Fig. 7e), except that these mRNAs are further stabilized by RINGmut. This increase in mRNA stability was validated for three selected IEG mRNAs by qPCR (Fig. 7f).

Minor comments:

1) On page 5, the authors list six proteins that they pull-down by endogenous FST-NOT1 purification. They mention that they did not further pursue CAPZA1 and CAPZAB. As a reader, it would be great if the authors can provide more information on why these two have not been further pursued, particularly because other factors may affect RNF219 activity on deadenylation.

> *Response: We now validated binding of the two actin-capping proteins CAPZA1 and CAPZB to the complex by FST-NOT1 IP and western blotting using specific antibodies (new panels in Fig. 1c). Indeed, the result shows that both proteins co-IP with FST-NOT1. Hence, CAPZA1 and CAPZB can be considered novel interactors of the CCR4-NOT complex, in*

addition to RNF219 and FHL2. As observed for FHL2, binding of CAPZA1 and CAPZB to the complex is not regulated by acetylation (additional quantification in Fig. 1d, e).

We would like to thank the reviewer for this comment as it helped us make better use of our proteomics results.

2) The authors mention on page 8 that RNF219- Δ SLiM shows only a small shift to the lighter fractions compared to WT because of its association to the origin recognition complex. Could the authors verify this by blotting for one of the proteins of the origin recognition complex?

> Response: To verify whether the ORC complex co-migrates with RNF219 in heavier fractions of the glycerol gradient we repeated the fractionation experiment shown in Fig. 2e and analyzed the co-sedimentation of ORC2 which is known to form a stable sub-complex with ORC3-5. As shown in Fig. 2e, RNF219- Δ SLiM co-sediments in lighter fractions of the glycerol gradient as compared to its WT counterpart, showing a small shift by one fraction (Fig. 2e and Reviewer Fig. 3 below). Whereas RNF219 and components of the CCR4-NOT complex co-sediment in fractions 7-14, the majority of ORC2 elutes in fractions 5 and 6 (Reviewer Fig. 3). This result excludes that a residual interaction with the ORC accounts for the small shift observed with RNF219- Δ SLiM. Based on this result we omitted our earlier statement and rephrased the text (line 212/213).

Since the minimal RNF219 construct (residues 434-600) is not as potent at inhibiting deadenylation as RNF219-C (residues 434-726; new Fig. 6c, d), an alternative explanation for the small shift observed with RNF219- Δ SLiM could be that additional regions within RNF219 may contribute to CCR4-NOT binding through low affinity interactions, which might be partially maintained in the glycerol gradient but not in the co-IP conditions.

3) Figure 2c: Listing the corresponding CNOT1 fragment in the sub-complexes used for pull-down would be beneficial for the understanding of the figure. Generally, the figure could be improved by labelling each protein in the pull-down.

> Response: In the revised Fig. 2c, all proteins have been labelled. The amino acid residues of the NOT1 fragments in each sub-complex have been included as well.

4) Could the authors clarify how many repetitions were done for the quantification of the glycerol gradient in Supp. Fig. 3b

> Response: The quantification shown in Supplemental Fig. 3b is based on one representative experiment (n=1). This information has been included in the figure legend (line 97).

5) RNF219 E3 ligase activity seems to be strikingly stimulated after incorporation into CCR4-NOT (Fig 3c-d). Is it possible to quantify the fold activation?

> *Response: In the two in vitro ubiquitination experiments, RNF219 is enriched by different means (IP of 3xFS-RNF219 from transiently transfected HEK cells in Supplementary Fig. 6b; IP of FST-NOT1 from endogenously tagged HeLa cells in Fig. 3c). Hence, one cannot directly compare the RNF219 autoubiquitination activity in the two experimental setups.*

For a direct comparison of the RNF219 autoubiquitination activity in the presence and absence of the CCR4-NOT complex, we now generated HeLa control and CNOT9 knock-out cells stably expressing equal levels of 3xFS-RNF219 (new Fig. 3d). The CCR4-NOT complex was co-purified with 3xFS-RNF219 in the control cells, but not in CNOT9 knock-out cells. This allowed us to directly compare the autoubiquitination activity of RNF219 in the presence and absence of the CCR4-NOT complex, and the result showed no difference in the degree of autoubiquitination between the two conditions (new Fig. 3d). Hence, RNF219 is an active E3 ligase when stably associated with the CCR4-NOT complex, yet its activity is not enhanced (or repressed) by the CCR4-NOT complex. We replaced former Fig. 3c by this more conclusive experiment (now Fig. 3d) and rephrased our interpretation of the E3 ligase activity of RNF219 (line 252/253).

6) Could the authors also add a quantification of their glycerol gradients in Supp. Fig 8 & 10?

> *Response: This comment now pertains to Supplementary Fig. 10 and 13. Quantification of the glycerol gradient distributions has been added in panels b.*

7) Please state the concentration of RNF219 and CNOT used in the ubiquitination assays.

> *Response: RNF219 and the CCR4-NOT complex were immuno-purified from transiently transfected HEK293 cells (Supplementary Fig. 6b) or from HeLa cells expressing endogenously tagged NOT1 (Fig. 3c). Following 3xFLAG peptide elution, equal quantities of the immunoprecipitated material were used for the +/- ATP condition. However, measuring absolute protein quantities is very difficult in IP eluates. Therefore, we do not know the concentration of RNF219 in the autoubiquitination assays, and there is no reason to assume that this information would affect our conclusions on the E3 ligase activity of RNF219.*

8) Top of page 9 – I presume this statement refers to the RNF219 SLiM and not all SLiMs that interact with NOT9? Is there any sequence conservation of RNF219 SLiM with other Caf40-binding motifs?

> *Response: Whilst there appears to be little conservation on the primary sequence level between the RNF219 SLiM and other experimentally verified Caf40-binding motifs (CBMs), our structural modeling and comparative analysis here suggests that a triad of hydrophobic residues in the C-terminal portion of the alpha-helical SLiM/CBM, which contact and engage a positively charged pocket on the surface of NOT9, are indeed conserved between species and in unrelated proteins (new Fig. 5b and new Supplementary 12b, c). Indeed, we confirmed that these three hydrophobic residues are essential for RNF219 to stably associate with the CCR4-NOT and to inhibit its deadenylase activity (new Fig. 6b, c,*

d), analogous to what was previously described for the Bag-of-marbles protein (<https://doi.org/10.1261/rna.064584.117>, <https://doi.org/10.1038/s41467-019-11094-z>).

9) Figure S2c – what are the two bands in the northern blot?

> Response: The two bands represent 28S and 18S rRNA - we now labeled the bands next to the poly(A) northern blot in Supplementary Fig. 2c. While ribosomal RNA is not considered to contain poly(A) tails, the specificity of these signals was confirmed using an oligo-dT/RNase H approach in our previous publication (<https://doi.org/10.1016/j.molcel.2016.08.030>, herein Supplemental Figure S1D). We assume that this represents a subpopulation of rRNA that contains poly(A) tails, maybe as part of a cellular rRNA quality control system.

Reviewer #3 (Remarks to the Author):

The evolutionarily conserved CCR4-NOT complex plays a vital role in controlling mRNA turnover and other aspects of mRNA metabolism both at the transcriptome level and in a transcript-specific manner. However, still little is known about how the complex is regulated and by what post-translational modifications and auxiliary factors. The Stoecklin lab showed a few years ago that protein acetylation has a profound impact on global mRNA turnover. The group showed that upon histone deacetylase (HDAC) inhibition global mRNA degradation is accelerated through CCR4-NOT complex. In this study, the authors set out addressing several related and critical issues by asking whether there may have acetylation-regulated changes in the protein interactome of the CCR4-NOT complex as well as physiological changes in the composition of the CCR4-NOT complex. Through well controlled mass-spec based purification and quantification experiments, they were able to pinpoint RNF219 as a potential auxiliary factor whose level diminishes upon RMD-elicited hyperacetylation. This in turn resulted in RNF219 diminished presence in the CCR4-NOT complex. Through a series of molecular and biochemical analysis, they identified the RNF219 interacting partner in the CCR4-NOT core complex, i.e., NOT9, and further mapped the subregions in RNF219 and NOT9 necessary for the interaction. After confirming the E3 ubiquitin ligase autoubiquitination activity, they showed that RNF219-mediated ubiquitination activity does not influence the composition of the CCR4-NOT complex or promote proteolytic degradation of core subunits. They then investigated whether RNF219 may actually modulate the deadenylation function of CCR4-NOT complex by conducting both in vitro and in vivo deadenylation assays. To determine transcriptome-wide impact of RNF219 on global mRNA decay, they generated RNF219 KO cell lines and brilliantly the isogenic lines that express either 3XFS-RNF wt or mutant for rescue experiments. These experiments showed that RNF219 protein per se and not its E3 ligase activity is required to inhibit the deadenylation function of CCR4-NOT complex and that RMD treatment induces degradation of RNF219 to lessen the inhibition.

Overall, this is a very interesting study, which reveals a novel mechanism by which CCR4-NOT complex is regulated through RNF219 and how HDAC inhibition elicits RNF219 degradation, thus activating the CCR4-NOT complex for deadenylation. The manuscript is well written with major conclusions properly drawn. The experiments and designs are well controlled and thought. The data are of strong quality. In this reviewer's opinion, all the data go into making an important story that is of broad and significant interest to general readers served by Nature Communications.

We thank the Reviewer for his/her encouraging assessment of our manuscript.

This reviewer has the following points for the authors' attention:

1. Lines 259-261: The authors showed that RNF219-C elicits a three- to four-fold reduction in the apparent deadenylation rate of the CCR4-NOT_{MINI} complex in vitro. They then concluded that "RNF219 acts as a potent inhibitor of CCR4-NOT-mediated deadenylation in vitro". This conclusion should be toned down as the in vivo demonstration that complements the in vitro results has not yet shown at this point. Alternatively, the authors may consider testing the intact RNF219 instead of the C-terminal portion in the experiment described in Fig. 4b

> Response: We rephrased this sentence now stating that "RNF219 has the capacity to directly inhibit CCR4-NOT-mediated deadenylation in vitro" (line 297/298). Since full-length RNF219 is not soluble when expressed recombinantly either in bacteria or in insect cells, attempts to purify this protein were so far not successful.

2. Lines 303-305: The authors concluded that "RNF219 is a general inhibitor of the CCR4-NOT deadenylase complex and this effect may be more pronounced for short-lived transcripts". The second statement would be more convincing if the authors can show the comparisons of stability changes between KO and WT cells in subgroups with different half-life ranges, e.g., transcripts with half-lives <6h (short-lived), transcripts with half-lives between 6h-12h (somewhat unstable/stable), and transcripts with half-lives >12h (long-lived) and that the short-lived subgroup is indeed affected the most among the three subgroups in the comparisons.

> Response: We followed the reviewer's advice and compared mRNA stability changes between parental HeLa, RNF219 KO, RNF219-WT rescue and RNF219-RINGmut rescue cells after grouping mRNAs into the following half-life intervals: $t_{1/2} \leq 3$ h, 3 h $< t_{1/2} \leq 6$ h, 6 h $< t_{1/2} \leq 9$ h, 9 h $< t_{1/2} \leq 12$ h, $t_{1/2} > 12$ h (new Fig. 7d). As expected, the first subgroup representing mRNAs with half-lives ≤ 3 h shows pronounced similarity with IEG mRNAs (Fig. 7e), since IEGs mostly comprise short-lived transcripts. To our surprise, the difference in mRNA stability between parental HeLa and RNF219 KO increases with increasing mRNA half-life, indicating that medium-to-long-lived transcripts are more affected by the loss of RNF219 expression than short-lived mRNAs.

Hence, we had to reconsider our model regarding the strength and target range of the RNF219 effect, and we removed our earlier statement about the preferential stabilization of short-lived mRNAs by RNF219. Another interesting feature revealed by the subgroup analysis is that RNF219-RINGmut loses its potency to rescue the RNF219 KO phenotype with increasing mRNA half-life (new Fig. 7d). This observation suggests that for short-lived mRNAs, RNF219-mediated inhibition of deadenylation and decay is independent of its E3 ligase activity, whereas stabilization of long-lived transcripts requires the integrity of its N-terminal RING domain.

We would like to thank the reviewer for proposing the subgroup analysis as it considerably improved the interpretation of our data and helped to tease out a regulatory difference between short- and long-lived mRNAs with regard to the RING domain of RNF219.

3. Supplementary Fig. 11b-d, transcripts used in each of these boxplots of half-life calculations for YTHDF2, TTP and Roquin proteins should be given as a supplementary table and references should be cited as well.

> Response: This comment now pertains to Supplementary Fig. 14 c-e. The transcripts were now compiled in Supplementary Table S3, and references were added to the figure legend (lines 424, 426, 427).

4. Line 324, "...the BTG/TOB family, which recruit mRNAs by associating with the cytoplasmic poly(A)-binding protein PABPC11,50,72." The TOB work (ref 54) should be cited here as well. The three references cited are only about BTG proteins that lack the typical PAM2 motif for PABP interaction.

> Response: Reference 54 has been added to this sentence (line 408).

5. In the second paragraph of Discussion (lines 331-344), the authors discussed the findings concerning RNF219 by other studies. The authors may consider mentioning that RNF219 was also shown by Chen et al. (RNA 26: 1143, 2020; see Supplemental Table S3) to be among the high-confidence proteins co-purified along with CCR4-NOT complex when using TOB2, a strong CAF1-interacting protein, as the bait. One possibility is that TOB2 may somehow counteract the inhibitory effect of RNF219 when joining the CCR4-NOT-RNF219 complex, given the close proximity of the NOT9 module to the CAF1-catalytic module which TOB2 binds to strongly.

> Response: We now included this reference and added a statement to the discussion on co-purification of RNF219 with the CCR4-NOT complex by TOB2 IP (line 421/422).

Reviewers' Comments:

Reviewer #1:

Remarks to the Author:

The authors have satisfactorily addressed most of the concerns I have raised. Now, the manuscript is substantially improved.

I have only some minor, technical issues left to be addressed as follows.

1. As the authors provided the mass spectrometry source data, I can spot Protein IDs starting with CON_ or REV_ in Supplementary Table S1. These are each potential contaminants or decoy (fake) protein groups, respectively. Did the authors remove these protein groups prior to analysis (which, in principle, should be)? I can't find the related analytic description in the text. If not, I strongly recommend the changes be made to Supplementary Table S1 and the corresponding figure (Fig. 1b).

2. In the 'Reporting for specific materials, systems and methods' section of the Reporting Summary file, the authors should describe the Antibodies and Eukaryotic cell lines used in this study.

Reviewer #2:

Remarks to the Author:

I am satisfied that the authors have address all of my comments. They included new experiments, validation and discussion. Overall this improved the manuscript substantially. Specifically, I appreciated how they validated the other interactions (CAPZA1, CAPZB), investigated RNF219 function with a CAF1 acetylation-deficient mutant, performed a more detailed analysis of RNA stability, and used AlphaFold to map the interaction site of RNF219 followed by mutational analysis. I have no further concerns

Reviewer #3:

Remarks to the Author:

The authors did an excellent and thorough job in revising the manuscript by experimentally addressing reviewers' major concerns. The additional data further strengthens their conclusions and also clarifies initial concerns raised by reviewers. The manuscript presents an important piece of work on a very interesting topic and in this reviewer's opinion is suitable for publication in Nature Communications.

Response to reviewers:

Reviewer #1:

Reviewer #1 (Remarks to the Author):

The authors have satisfactorily addressed most of the concerns I have raised. Now, the manuscript is substantially improved.

I have only some minor, technical issues left to be addressed as follows.

1. As the authors provided the mass spectrometry source data, I can spot Protein IDs starting with CON_ or REV_ in Supplementary Table S1. These are each potential contaminants or decoy (fake) protein groups, respectively. Did the authors remove these protein groups prior to analysis (which, in principle, should be)? I can't find the related analytic description in the text. If not, I strongly recommend the changes be made to Supplementary Table S1 and the corresponding figure (Fig. 1b).

> Response: We thank the reviewer for pointing out this apparent discrepancy. Supplementary Table S1 contains MaxQuant-derived raw proteomics data from four independent FST-NOT1 purifications using DMSO- or RMD-treated FST-NOT1 or parental HeLa cells as control. Hence, this list still contains potential contaminants and decoy protein groups. Downstream data analysis was performed using LFQ values in Perseus v1.6.2.1. First, commonly occurring contaminants, proteins only identified by a modification site or those matching the reversed part of the decoy database were excluded. Second, proteins were filtered for LFQ values being present in at least 3 samples across the four biological repeat experiments. Means of log₂-transformed LFQ values were calculated and core CCR4-NOT subunits as well as candidate interactors were revealed based on their exclusive identification in FST-NOT1 HeLa cells (visualized in Fig. 1b). To allow for an unbiased evaluation of our proteomics dataset, we decided to provide an unprocessed and unfiltered ProteinGroup file as Supplementary Table S1 (now Supplementary Data 1). We have now added a detailed description of the proteomics data analysis to the methods section under "Analysis of mass spectrometry data". The filtered list of FST-NOT1-associated proteins is now included in the Source Data file.

2. In the 'Reporting for specific materials, systems and methods' section of the Reporting Summary file, the authors should describe the Antibodies and Eukaryotic cell lines used in this study.

> Response: We now added more detailed information concerning the identity, commercial source, and validation of antibodies to the Reporting Summary accompanying this manuscript. The source of eukaryotic cell lines and the method used for confirming their authenticity has been included as well.

Reviewer #2 (Remarks to the Author):

I am satisfied that the authors have address all of my comments. They included new experiments, validation and discussion. Overall this improved the manuscript substantially. Specifically, I appreciated how they validated the other interactions (CAPZA1, CAPZB), investigated RNF219 function with a CAF1 acetylation-deficient mutant, performed a more

detailed analysis of RNA stability, and used AlphaFold to map the interaction site of RNF219 followed by mutational analysis. I have no further concerns

> Response: We thank the reviewer for his/her constructive criticism and suggestions which prompted further investigations that lead to a substantial improvement of our manuscript.

Reviewer #3 (Remarks to the Author):

The authors did an excellent and thorough job in revising the manuscript by experimentally addressing reviewers' major concerns. The additional data further strengthens their conclusions and also clarifies initial concerns raised by reviewers. The manuscript presents an important piece of work on a very interesting topic and in this reviewer's opinion is suitable for publication in Nature Communications.

> Response: We appreciate the reviewer's positive feedback on our revised manuscript.